# RECURSION OF THOUGHT: DIVIDE AND CONQUER REASONING WITH LANGUAGE MODELS

## ABSTRACT

With the recent advances in language models, attempts are being made to apply them to solving multi-step reasoning problems. A major breakthrough in this line of research is to let language models generate intermediate steps, often called Chain of Thought (CoT), before producing a final answer. However, language models have an upper bound on the context size, i.e., the number of input tokens, such as 2048 for the recent GPT-3 and PaLM. Although several thousand tokens are enough to handle various tasks, solving more complex reasoning tasks can require orders of magnitude more tokens. Therefore, the context limit imposes a fundamental limit on the model's reasoning capability. Inspired by human's incredible reasoning ability based on abstraction and recursion, we propose *Recursion of Thought* (RoT) as a model-agnostic framework with the novel paradigm of teaching a language model to divide and conquer complex problems by recursively creating multiple contexts. Since RoT casts the context-related operations as tokens, a language model can trigger the recursion operations by simply producing the corresponding tokens. On multiple arithmetic and algorithmic reasoning tasks, we demonstrate that RoT dramatically improves the recent large-scale language model GPT-3 to solve extremely complex problems. Moreover, RoT can make tiny, randomly initialized Transformers or LSTMs to solve problems that even humans find daunting.

## 1 INTRODUCTION

Recently, language models (LMs) have become a prominant direction to solve reasoning. Given a question sequence, the models are tasked to predict the following answer sequence. One recent line of research for reasoning with LMs is *chain of thought* (CoT) generation (Nye et al., 2021; Wei et al., 2022; Kojima et al., 2022; Lewkowycz et al., 2022). In CoT generation, complex reasoning problems are solved by generating intermediate reasoning steps, or a chain of thought, before producing the final answer. Directly answering a question would require a model to fully solve the problem in a single forward pass, meaning the range of solvable problems is severely limited by the model's capacity. On the other hand, generating CoT before the answer allows the problem's complexity to be spread across the CoT, making each token generation more straightforward given the previous tokens. This is closer to how humans solve complex problems, as we think step by step, instead of producing an answer reflexively.

Although CoT seems promising, there is a critical issue that significantly limits its utility: the effective context size of sequence models cannot grow unbounded. In this work, context refers to the set of input tokens that a model is conditioned on when generating output. Practically, all sequence models have a limit on the maximum context length due to various reasons. For instance, Transformers (Vaswani et al., 2017) suffer from a quadratic computational cost on the context length, and RNNs (Hochreiter & Schmidhuber, 1997) struggle with long-term dependency modeling. Therefore, even the state-of-the-art language models, such as GPT-3 (Brown et al., 2020) and PaLM (Chowdhery et al., 2022), limit the maximum context length by up to 2048 tokens. However, the length of intermediate steps can grow rapidly with the problem's complexity and exceeds the context limit. Since CoT can handle a problem only if the process of solving it fits into a single context, the range of problems that CoT can handle is severely constrained by the context limit. This issue must be

overcome to solve more challenging and useful reasoning problems, whose solutions may require millions of tokens.

Humans can handle this issue by using abstraction and recursion. We divide a large problem into smaller subproblems and focus on each subproblem while solving it, instead of considering the entire problem at every step. We can further subdivide a subproblem into even smaller subproblems. With this intuition, we propose *Recursion of Thought* (RoT) as a model-agnostic framework for recursively solving multi-step reasoning problems. The key feature of RoT is to grant the model the ability to recursively create and utilize multiple contexts for subproblems. We achieve this feat by introducing several special tokens that a model can output to control its context. During inference, the model recursively solves the problems by producing appropriate tokens at the right time. Moreover, RoT supports tail recursion, which enables general computation with an indefinitely long chain of recursion.

We demonstrate RoT on four basic arithmetic operations (addition, subtraction, multiplication, and division) and four algorithmic tasks (longest common subsequence, longest palindromic subsequence, 0-1 knapsack, and matrix chain multiplication) to show its generality. Without any task-specific component, such as a calculator, all tasks are formulated as autoregressive sequence modeling problems. These tasks require a model to generalize by just seeing a tiny fraction of the problem space since the space is combinatorially large. For example, even in simple arithmetic operations, two 6-digit operands result in one trillion possible combinations. Hence, we evaluate whether a model understands the underlying rules, instead of brute force memorization. In our experiments, the range of problems that CoT can handle is seriously constrained by the context limit. On the other hand, RoT leads language models to achieve near perfect accuracy, even if the problem size increases to the extreme, where solving one problem requires producing hundreds of thousands of tokens. Moreover, the dramatic improvement is not limited to large pre-trained language models like GPT-3. RoT can make tiny, randomly initialized Transformers or LSTMs perform extremely complex reasoning.

The key messages of this work are summarized as follows:

- The reasoning capability of current language models is seriously constrained by the maximum length of a *single context*.

- Our *Recursion of Thought* (RoT) unleashes the reasoning capability of language models by letting them recursively create and utilize *multiple contexts* of subproblems, following the principle of divide and conquer.

In the supplementary file, we provide the source code to *fully* reproduce our experiments.

## 2 RELATED WORK

**Chain of Thought.** Among several prior works on applying language models to reasoning, Scratchpad (Nye et al., 2021) may be the most closely related to our work. It is the first approach to fine-tune language models to produce CoT before generating an answer. It demonstrates its effectiveness on 8-digit addition, polynomial evaluation, and Python program execution. It also mentions the confined context size as a major limitation to be overcome. In order to unlock the full potential of Scratchpad, the authors argue that Transformers should be improved to allow greater context sizes. We solve this exact problem from a completely different perspective, i.e., using multiple contexts to divide-and-conquer. Our approach is more practical and scalable, compared to increasing the context limit. More recently, it has been found that sufficiently large pre-trained language models can be induced to produce CoT, by simply tuning the *prompt*. For instance, CoT prompting (Wei et al., 2022) adds several QA exemplars with CoT before the main question, encouraging the model to generate final answers in the similar manner. Kojima et al. (2022)'s prompting is even simpler; after a question, they start the answer with "Let's think step by step," and then let the model finish the rest. Even without fine-tuning, these methods significantly improve the reasoning accuracy of language models. Minerva (Lewkowycz et al., 2022) utilizes these prompting techniques with a specially curated scientific pre-training dataset to achieve remarkable results on various reasoning benchmarks. However, all of these works are still limited by the maximum context size.

**Neural Programmer-Interpreter (NPI).** Unlike language models, NPI (Reed & de Freitas, 2016) interacts with its environment through a series of program execution. It consists of an LSTM core, an encoder for each domain, and a memory of program embeddings. At every time step, the LSTM core takes a program embedding, arguments, and an observation of its environment to produce the next program embedding and corresponding arguments. Cai et al. (2017) combine NPI with recursion and show that recursion plays a critical role in generalization. Since NPI requires full execution traces for training, there are multiple works to relax this requirement using reinforcement learning (Li et al., 2017; Fox et al., 2018; Pierrot et al., 2019).

**System 1 Approaches for Reasoning.** Kahneman (2013) classifies cognitive tasks into two categories: System 1 and System 2. System 1 refers to fast and reflexive thinking, while System 2 refers to sequential reasoning. It is hard to define strict criteria to distinguish between System 1 and System 2 approaches. We classify a model as System 1 if it directly outputs an answer, while System 2 generates its process, as well. In that sense, RoT, CoT and NPI are System 2 approaches, while there have been various System 1 approaches to solve symbolic reasoning. Zaremba & Sutskever (2014) train LSTMs with curriculum learning to solve integer addition up to nine digits. Kaiser & Sutskever (2016) propose a convolutional architecture called Neural GPU that performs binary addition and multiplication. Trained on 20-bit problems, this model operates like a digital circuit for the arithmetic operations, which can generalize up to 2,000 bits. Similarly, Yan et al. (2020) solve 8-bit binary addition and 12-bit binary multiplication with Transformers. If Neural GPUs are like digital circuits, Neural Arithmetic Logic Units (Trask et al., 2018) are like analog circuits for arithmetic. They represent numerical quantities with the activation values of neural networks and design a clever architecture for arithmetic operations. Although these System 1 approaches fall behind System 2 methods in terms of generality, they can be highly efficient in specific domains. We believe future systems will often be a hybrid of System 1 and System 2, harnessing the advantages of both types.

## 3 Recursion of Thought

The main idea of *Recursion of Thought* (RoT) is to let the model recursively solve small subproblems in separate contexts, keeping each reasoning step simple and learnable. Our RoT is model-agnostic and general enough to be combined with any kind of sequence model that supports autoregressive generation. The only requirement is that the model should be able to infer $p(x_{i+1}|X_{1:i})$, the probability of the next token $x_{i+1}$ given a sequence $X_{1:i} = [x_1; ...; x_i]$. Therefore, sequence models such as Transformers, RNNs, or more advanced ones can all be used in the RoT framework. RoT teaches a sequence model to solve a problem using the divide and conquer paradigm in a supervised manner. That is, we assume that ground truths for the intermediate steps of how to recurse are readily available, as in Scratchpad (Nye et al., 2021) or NPI (Reed & de Freitas, 2016).

For better understanding, we discuss RoT in the reverse order of the pipeline. In §3.1, we first describe how to perform RoT inference with a fully trained model. In §3.2, we introduce the training process. Finally, in §3.3, we discuss how to recursively divide the problems and build the training data for intermediate steps automatically.

### 3.1 Inference

Recursion of Thought (RoT) grants a language model the ability to control the recursion process. For basic recursion control, we first introduce the following special tokens: `GO`, `STOP`, and `THINK`. `GO` and `STOP` respectively mark the start and end of a problem sequence. They can be nested inside another `GO`-`STOP` pair to indicate a subproblem. `THINK` initiates a recursion procedure. RoT teaches a model how to use these tokens so that it can perform divide-and-conquer problem solving. We formulate each inference context of a QA problem, denoted $X$, as the following concatenation:

$$X = [Q; Q^{\text{sub},1}; A^{\text{sub},1}; \ldots; Q^{\text{sub},N}; A^{\text{sub},N}; A] \tag{1}$$

where $Q$ and $A$ are the main question and answer sequence, and $Q^{\text{sub},*}$ and $A^{\text{sub},*}$ are the questions and answers of the *top-level* subproblems. During inference, a model is given $Q$ and tasked to generate the rest. Questions (both $Q$ and $Q^{\text{sub},*}$) start with a `GO` token, and answers (both $A$ and $A^{\text{sub},*}$) end with a `STOP` token. For trivial cases, i.e., the base cases of recursion, the context

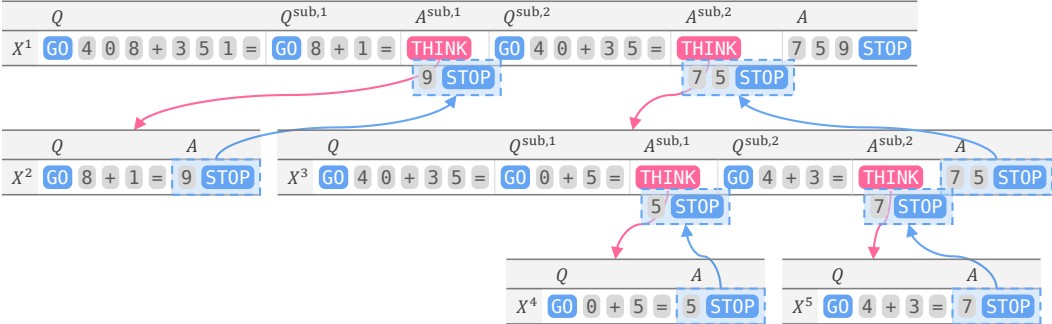

Figure 1: An example of the Recursion of Thought inference in Algorithm 1. Each table represents an inference context $X^k$ in order of creation, which has the structure of Eq.1. For each context, the model is given $Q$ and tasked to generate the rest, one token at a time. The model outputs the THINK token when it needs to generate $A^{\text{sub},*}$, the answer of a subproblem. The THINK token triggers a recursive process that solves the subproblem in a new context and returns the answer.

| $Q$ | $Q^{\text{sub},1}$ | $A^{\text{sub},1}$ | $Q^{\text{sub},2}$ | $A^{\text{sub},2}$ | $A$ |
|---|---|---|---|---|---|
| $X^1$ `GO` 4 0 8 + 3 5 1 = | `GO` 8 + 1 = 9 | `STOP` | `GO` 4 0 + 3 5 = 7 | 5 `STOP` | 7 5 9 `STOP` |
| $Y^1$ `PAD` $\cdots$ `PAD` | `GO` 8 + 1 = `THINK` `PAD` | | `GO` 4 0 + 3 5 = `THINK` `PAD` `PAD` | | 7 5 9 `STOP` |

Figure 2: The target sequence $Y^1$ is produced from $X^1$ in Figure 1 by Algorithm 2. Given $X^1_{1:i}$, the model is trained to output $Y^1_{i+1}$, except for `PAD`s that are ignored. Note `PAD` is a dummy token to make sure $\|X\| = \|Y\|$.

contains no $(Q^{\text{sub},*}, A^{\text{sub},*})$ pair. A subproblem can have smaller, lower-level subproblems recursively, but only the top-level subproblems remain in a context. As a result, we can abstract away the details for solving the subproblems and keep only the high-level results in the current context.

For tail recursion, where the last subquestion's answer becomes the final answer, we additionally introduce the TAIL token. If TAIL is used in the place of a GO token in the last subquestion $Q^{\text{sub},N}$, its answer $A^{\text{sub},N}$ is treated as the final answer $A$, and the context $X$ does not have duplicate $A$.

Algorithm 1 summarizes the inference process. Figure 1 presents a example of solving $408 + 351$ for better understanding. More detailed illustrations of inference can be found in Appendix A. The RoT function (L1) takes a fully trained model with a question $Q$ as input and returns the answer $A$ as output. The procedure starts by initializing the context $X$ with the original question $Q$ (e.g., GO 4 0 8 + 3 5 1 = in Figure 1). $i_{\text{ans}}$ is the starting index of the answer, which is initialized to $|X| + 1$, where $|\cdot|$ is the length of a sequence. Then, in the main loop, the model iteratively generates a next token $x$ from $X$, which is appended to the end of $X$. After the initialization, the model is expected to (i) generate answer $A$ directly or (ii) output GO. If the model immediately generates $A$ and finishes it with a STOP, the answer is returned (L9), which is the base case of the recursion. Up to this point, the algorithm is identical to common language models.

On the other hand, if the model decides to output GO, which signals the start of a subproblem, its index is stored in $i_{\text{go}}$ (L11). Since a subproblem has started, the next several tokens from the model should constitute the question of the subproblem. In Figure 1, the first subproblem of the main context $X^1$ is adding the last digits, i.e., $8 + 1$. Once the subquestion is generated, the next step is to find an answer to it. This is the pivotal moment: *instead of producing the answer, the model outputs* the THINK token, which initiates the recursion with a new context (L16-23). First, we separate the subquestion $Q^{\text{sub}}$, starting from $i_{\text{go}}$ (L16). Second, using it as an input, we trigger the recursive call (L17) to obtain the answer $A^{\text{sub}}$. As shown as red arrows in Figure 1, this call creates another context $X^2$ and initializes its question part with the subquestion. Then, the same inference process is executed inside the new context, sometimes running more recursions. In the case of $8 + 1$, the answer 9 STOP is immediately returned since it is a base case. Finally, the THINK token is replaced

with $A^{\text{sub}}$ (L21) and the starting position of the answer is updated (L23). Hence, when predicting the next token, the model sees the returned answer as input, instead of the THINK token.

Once a subproblem is finished, the model can solve another subproblem in the same way, or output the final answer. In Figure 1, the second subproblem is to add all the remaining digits, i.e., $40 + 35$. To solve it, the model recursively solves two other subproblems, i.e., adding each digit, before generating the answer. If the model outputs the final answer followed by a STOP token, the answer is returned (L9). If the model starts a subproblem with TAIL instead of GO, it becomes a tail recursion (L14), and its answer is returned directly as the final answer (L19).

### 3.2 TRAINING

We teach RoT in a supervised manner; the model is trained with the ground truth (GT) intermediate steps, which also include when to output the special tokens. Each training example is constructed as a pair of a ground truth context sequence $X$ and the corresponding target sequence $Y$. The GT context $X$ is structured as Equation 1 and automatically built by the algorithms that will be introduced in §3.3 and Appendix D. In this section, we discuss how to construct the target sequence $Y$ for $X$, and define the training objective.

Algorithm 2 summarizes the process of converting $X$ to $Y$, where $Y$ has the same length with $X$. Refer to Figure 2 for an example. Overall, $Y$ is a copy of $X$ except for the parts corresponding to $Q$ and $A^{\text{sub},*}$. Since the question $Q$ is always given in a context, $Q$ is replaced by special PAD tokens (L1), which mean "nothing to predict for this part." Each subproblem's answer $A^{\text{sub},n}$ is replaced by a THINK token followed by several PAD s that fill in the rest to make sure $|X| = |Y|$ (L4). This way, the model is trained to output THINK instead of the first token of $A^{\text{sub},n}$. Since the whole $A^{\text{sub},n}$ will be returned from the recursive process and replace the THINK during inference (L17,21 of Algorithm 1), we do not need a training signal for the rest of $A^{\text{sub},n}$.

Given a pair $(X, Y)$, the training objective is defined as follows:

$$\mathcal{L} = -\sum_i I[y_{i+1} \neq \text{PAD}] \log p(y_{i+1}|X_{1:i}) \quad (2)$$

---

**Algorithm 1** Recursion of Thought Inference

**Require:** A sequence model $\mathcal{M}$ trained for Recursion of Thought, a question sequence $Q$
1: **function** RoT($\mathcal{M}$, $Q$)
2: $\quad$ $X \leftarrow Q$ $\quad$ ▷ Initialize context with $Q$
3: $\quad$ $i_{\text{ans}} \leftarrow |X| + 1$ $\quad$ ▷ Start of answer
4: $\quad$ $t \leftarrow false$ $\quad$ ▷ Tail recursion
5: $\quad$ **while** True **do**
6: $\quad\quad$ $x \leftarrow \mathcal{M}(X)$ ▷ Generate next token
7: $\quad\quad$ $X \leftarrow [X; x]$
8: $\quad\quad$ **if** $x = $ STOP **then**
9: $\quad\quad\quad$ **return** $X_{i_{\text{ans}}:|X|}$
10: $\quad\quad$ **else if** $x = $ GO **then**
11: $\quad\quad\quad$ $i_{\text{go}} \leftarrow |X|$ $\quad$ ▷ Mark last GO
12: $\quad\quad$ **else if** $x = $ TAIL **then**
13: $\quad\quad\quad$ $i_{\text{go}} \leftarrow |X|$
14: $\quad\quad\quad$ $t \leftarrow true$ ▷ Mark tail recursion
15: $\quad\quad$ **else if** $x = $ THINK **then**
16: $\quad\quad\quad$ $Q^{\text{sub}} \leftarrow X_{i_{\text{go}}:|X|-1}$
17: $\quad\quad\quad$ $A^{\text{sub}} \leftarrow $ RoT($\mathcal{M}$, $Q^{\text{sub}}$)
18: $\quad\quad\quad$ **if** $t$ **then**
19: $\quad\quad\quad\quad$ **return** $A^{\text{sub}}$
20: $\quad\quad\quad$ **end if**
21: $\quad\quad\quad$ $X \leftarrow [X_{1:|X|-1}; A^{\text{sub}}]$
22: $\quad\quad\quad$ $\quad$ ▷ Replace THINK with $A^{\text{sub}}$
23: $\quad\quad\quad$ $i_{\text{ans}} \leftarrow |X| + 1$
24: $\quad\quad$ **end if**
25: $\quad$ **end while**
26: **end function**

---

**Algorithm 2** Creating the target sequence

**Require:** Context $X = [Q; Q^{\text{sub},1}; A^{\text{sub},1}; \ldots; Q^{\text{sub},N}; A^{\text{sub},N}; A]$
1: $Y \leftarrow \underbrace{\text{PAD} \ldots \text{PAD}}_{|Q|}$
2: **for** $n$ in $1 \ldots N$ **do**
3: $\quad$ $Y \leftarrow [Y; Q^{\text{sub},n}]$
4: $\quad$ $Y \leftarrow [Y; \text{THINK} \underbrace{\text{PAD} \ldots \text{PAD}}_{|A^{\text{sub},n}|-1}]$
5: **end for**
6: $Y \leftarrow [Y; A]$
7: **return** $Y$

---

where $I$ is the indicator function that excludes PAD s from training. Simply, it means that the sequence model is trained to output $y_{i+1}$ as the next token for a given $X_{1:i}$. Its form is almost identical to the standard language modeling objective: $\mathcal{L}_{\text{LM}} = -\sum_i \log p(x_{i+1}|X_{1:i})$, which is to predict the next token given previous tokens. With this objective, any sequence model is trained in the standard way, i.e., end-to-end via stochastic gradient descent. For decoder-only transformers with causal masks, the training can be efficiently done in parallel for all tokens.

### 3.3 THE RECURSIVE REASONING PROCEDURES

As explained in §3.2, we teach the recursive reasoning procedure for a problem type by providing ground truth contexts. To produce these contexts, we design an exemplary reasoning procedure for each problem type and implement it as a *teacher* program that automatically generates the contexts given a problem. The major desiderata for the exemplary reasoning procedures are two fold: (i) each context should be short, and (ii) the next tokens to generate should be obvious from the previous sequence. Although the definition of "being obvious" may vary depending on the model's architecture (e.g., Transformers vs. LSTMs), the reasoning procedures developed for human generally meet these desiderata. Therefore, our procedures for the arithmetic problems are borrowed from the elementary school math. For example, the addition procedure shown in Figure 1 is a recursive version of adding digits one by one starting from the last digit. In another example of multiplication, an $N$-digit $\times$ $M$-digit multiplication is divided into an $N$-digit $\times$ 1-digit multiplication and an $N$-digit $\times$ $(M-1)$-digit multiplication. The $N$-digit $\times$ 1-digit multiplication is further divided into a 1-digit $\times$ 1-digit multiplication and an $(N-1)$-digit $\times$ 1-digit multiplication. For the algorithmic problems, we also borrow standard, well-known algorithms. In Appendix D, we provide the full details of the procedures for each problem type, with Python code snippets of the key parts. Note that our proposals for the reasoning procedures in Appendix D are one of many possible solutions, which are not necessarily optimal.

**Training Data Distribution.** We use the same problem distribution for both training and evaluation, since out-of-distribution generalization is not within the scope of this paper. That is, when teaching 6-digit multiplication to the model, both training and test sets are all examples of 6-digit multiplication. The problem distributions are elaborated in Appendix C. Another important detail regarding the training of RoT is that each training example in a batch is a context, not a whole problem. Since RoT generates multiple contexts per problem, often a large portion of contexts can be duplicate (mostly the base cases). Therefore, to build a training batch for RoT, we first sample a top level problem and find the set of unique RoT contexts from the problem. Out of the unique contexts, we randomly sample one context as a training example. We find this simple technique works well, and we do not need more sophisticated method, such as the adaptive curriculum learning in Reed & de Freitas (2016).

## 4 EXPERIMENTS

Since Recursion of Thought is the first approach of this kind, we mainly compare with two baselines. The first one is to output an answer directly from a question, which we call *Without Thought* (WT). The other one is to generate all the intermediate steps before the answer without recursion (Nye et al., 2021), which we refer to as *Chain of Thought* (CoT; not to be confused with the CoT *prompting* (Wei et al., 2022)) for consistency. We construct the ground truths for CoTs by unraveling the same recursive process which we design for RoT, into a single context sequence (see Appendix B for examples). Therefore, the number of tokens to generate while solving a problem is the same for both CoT and RoT (if we do not count the `THINK` tokens). However, the sizes of the individual contexts of CoT are far longer than those of RoT due to the recursively nested subproblems. Refer to Appendix I for more detailed analysis of the context sizes. Note that we *train* these baselines and do not use any prompting technique. When evaluating, we consider a problem to be correctly solved only if all the intermediate steps and the answer are correct. In other words, we impose stricter rules on both RoT and CoT by not counting "lucky guesses" as correct.

### 4.1 THE REASONING PROBLEMS

To evaluate the reasoning capabilities, we test various reasoning tasks that are grouped into two categories: arithmetic reasoning and algorithmic reasoning. We below provide a rough description of the tasks, whose details can be found in Appendix C. All the reasoning tasks share one characteristic in common: we can easily adjust the problem's difficulty. Therefore, we can gradually increase the degree of difficulty and see which method fails first. Since the goal of our experiments is to test the reasoning capability of language models, all problems are formulated in pure sequence modeling, with no external program (e.g., calculator) called by the models.

**Arithmetic Reasoning.** We test four basic arithmetic operations, i.e., addition, subtraction, multiplication, and division, with two non-negative integers. The difficulty of arithmetic problems is represented by the maximum number of digits in an operand. For instance, in 6-digit multiplication, each operand can range from 0 to 999,999. When we sample a problem, we sample each operand from the log-uniform distribution. Compared to the uniform distribution where the samples are highly biased towards extremely large numbers, we get roughly the same ratio of samples for each number of digits. The sampling schemes for each operation are elaborated in Appendix C.1.

**Algorithmic Reasoning.** We test four algorithmic tasks with distinct characteristics. These problems are generally solved via dynamic programming (DP), and the length of intermediate steps can increase rapidly since the time complexity of DP algorithms ranges from $\mathcal{O}(N^2)$ (LCS, LPS, and 0-1 Knapsack) to $\mathcal{O}(N^3)$ (MCM). (1) *Longest Common Subsequence* (LCS): Given two random sequences of length $N$, the model finds the longest common subsequence and its length. The sequences consist of 10 characters from 0 to 9, and the problem difficulty is defined to be the sequence length $N$. (2) *Longest Palindromic Subsequence* (LPS): Similar to LCS, given a random sequence of length $N$, the model finds the longest palindromic subsequence and its length. (3) *0-1 Knapsack*: Given a list of $N$ items with specific value and weight, the model finds the best combination of items that maximizes the total value under the weight limit of a knapsack. The problem difficulty is represented by the number of items. (4) *Matrix Chain Multiplication* (MCM): The computational cost of multiplying $N$ (> 2) matrices varies greatly depending on the order of multiplication. MCM is the task of finding the best multiplication order that yields minimal computation cost. The difficulty is controlled by the number of matrices $N$.

## 4.2 Unleashing GPT-3's Reasoning Capability through Recursion of Thought

Despite their remarkable language modeling capabilities, the state-of-the-art large language models, including GPT-3 (Brown et al., 2020), struggle to solve even the basic arithmetic tasks (Nye et al., 2021). For instance, it cannot correctly handle multiplication with more than one or two digits. Using the OpenAI API, we fine-tune GPT-3 on the reasoning tasks in §4.1 for 10K steps with a batch size of 256. Each training batch is randomly sampled from the training data distribution explained in §3.3. The results are presented in Figure 3a, and the technical details are described in Appendix E. Each point in the graphs represents one experiment at a certain problem difficulty. We report the accuracy on a test set of 1K unique problems randomly sampled as explained in Appendix C. To the best of our knowledge, the problems at this scale (e.g., 48-digit addition/subtraction and 16-digit multiplication/division) have never been solved by any language model without the help of external programs. For reference, Minerva (Lewkowycz et al., 2022) achieves around 80% accuracy on 10-digit addition and 20% on 18-digit addition.

Even WT fine-tuning cannot make GPT-3 deal with such a level of complexity, while CoT is not applicable due to the context limit of 2048. The green dotted lines mark the maximum difficulty that can be handled by CoT under the context limit. On the other hand, RoT finetunes the GPT-3 to achieve near perfect scores in every experiment. As presented in Appendix I, solving each problem requires up to tens of thousands of tokens. Without any architectural change, RoT makes GPT-3 handle these extremely complex problems.

## 4.3 Recursion of Thought with Tiny Language Models

Recent research on reasoning has been mostly focused on extremely large pre-trained language models. In this section, we show an interesting result that RoT can make even tiny models, without any pre-training, perform convoluted reasoning procedures. Since RoT is model-agnostic, we test the two basic sequence model architectures: Transformer Vaswani et al. (2017) and LSTM Hochreiter & Schmidhuber (1997). For Transformer, we use a decoder-only model with 4 layers, 2 attention heads, 128 embedding dimensions, and 256 feed-forward dimensions, a total of only 536K parameters. It is a million times smaller than the largest PaLM (Chowdhery et al., 2022) with 540B parameters. The context limit is set to 2048 following GPT-3 and PaLM. For LSTM, we use 4 layers, 64 input dimensions, and 256 hidden dimensions, which result in 272K parameters. We set the context limit of the LSTM to 512 since (i) it takes a lot of time for LSTMs to process the tokens sequentially, and (ii) they are not good at handling long-term dependency.

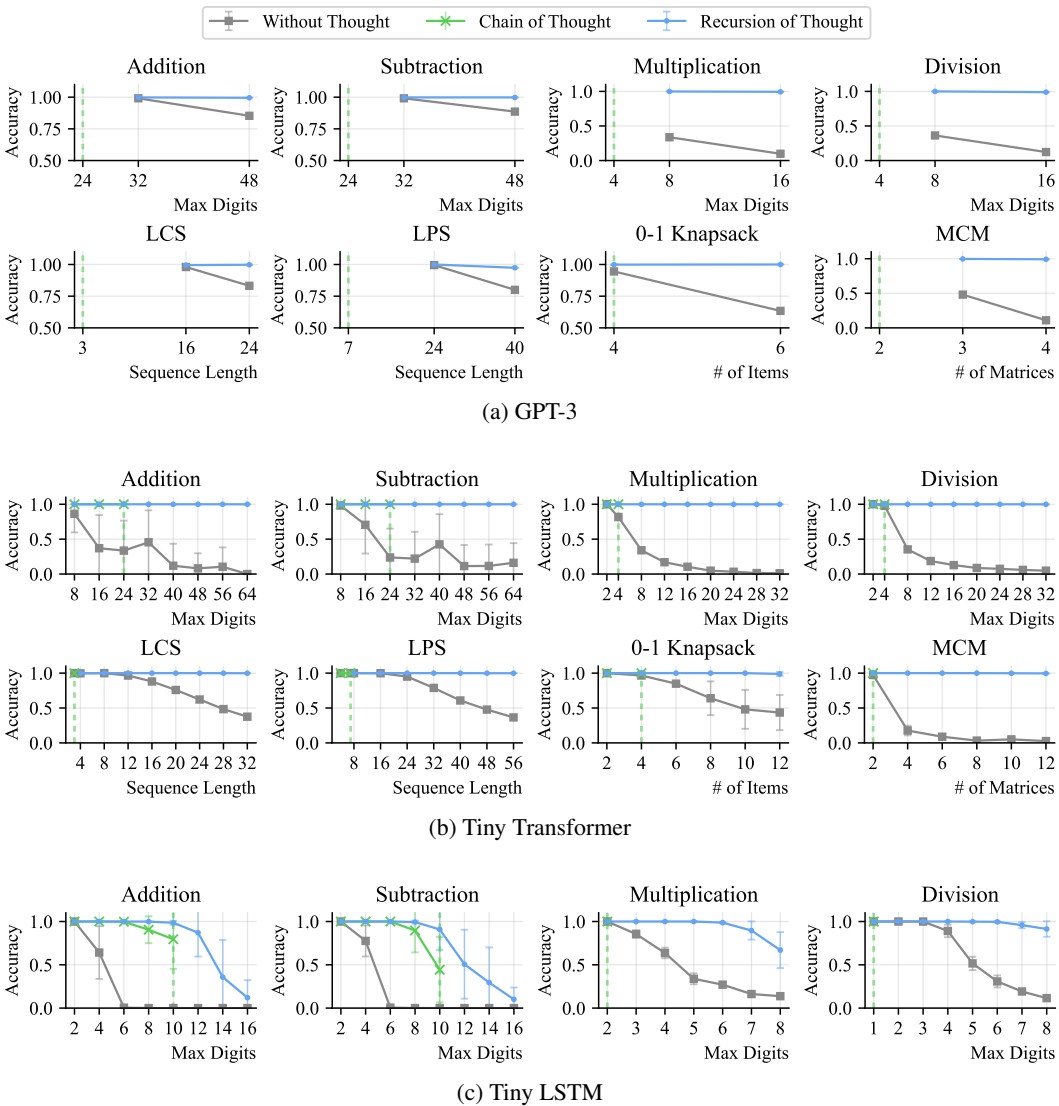

Figure 3: Comparison of the thought processes. In each graph, the x-axis is the problem difficulty, while the y-axis is the reasoning accuracy. Each point represents an independent experiment. The green vertical lines indicate the maximum problem difficulty that CoT can handle without exceeding the maximum context size.

By virtue of their small sizes, we conduct far more extensive experiments than GPT-3, which are presented in Figure 3b and Figure 3c. We test both arithmetic and algorithmic reasoning problems with Transformer, and the arithmetic problems with LSTM. For each experiment, we train a randomly initialized model and evaluate it on a test set of 30K unique problems. With a batch size of 256, Transformers and LSTMs are trained for 500K steps and 800K steps, respectively. We repeat each experiment eight times and report the average and standard deviation of the accuracies. Appendix K enumerates the exact values of Figure 3. With the tiny Transformer, we experiment to the extent where even humans would find daunting. For example, we test addition/subtraction up to 64 digits and multiplication/division up to 32 digits. Note that a 32-digit number cannot even fit into the 64-bit integer datatype.

Throughout the experiments, we observe consistent patterns:

- WT's accuracy drops most quickly as the problem difficulty increases.

- CoT achieves near perfect accuracy, but it can only be applied to simple problems due to the context limit.
- RoT achieves near perfect accuracy and can be scaled up to extremely complex problems.

Despite the small sizes, RoT makes the Transformers master all types of extremely complex problems. We do not test more difficult problems mainly because the evaluation becomes too costly, not because RoT is incapable of learning them.

## 5 DISCUSSION

The results of the tiny Transformer suggest that we might have to rethink the capability of large language models. If RoT enables the tiny Transformer to easily master 32-digit multiplication or division, what would a million times bigger model, like PaLM, be capable of? In contrast to the currently ongoing arms race in language models, the number of parameters might not be the main bottleneck anymore to increase models' reasoning capability. We believe that our new paradigm of utilizing multiple contexts has the potential to make a huge leap in this line of research.

The current limitation of RoT is the need for supervision to learn divide and conquer for each task. In order to apply RoT to a wider range of tasks, it may be crucial to reduce the expensive supervision. As one possible approach, we may borrow the RL-based methodologies that are developed for reducing supervision of NPI (Li et al., 2017; Fox et al., 2018; Pierrot et al., 2019).

Interestingly, RoT cannot facilitate length generalization, e.g., training on 8-digit multiplication with RoT cannot make a model generalize to 16-digit multiplication. We believe this problem is rooted in more fundamental limitation of the Transformer architecture (Hahn, 2020), orthogonal to RoT. Fortunately, since RoT is a model-agnostic framework, we would be able to apply RoT to more advanced architectures to come in the future, which might be capable of length generalization.

## 6 CONCLUSION

Despite the remarkable advances in language models, their reasoning capability has always been constrained by the maximum size of a single context. In this work, we introduce Recursion of Thought to solve this problem by utilizing multiple contexts. We prove its potential through extensive experiments, showing that it is possible to make language models solve problems that require hundreds of thousands of tokens. We believe the core idea of utilizing multiple contexts will play an essential role in future language models.

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

## A  A STEP BY STEP ILLUSTRATION OF RoT INFERENCE

In this section, we provide a step by step illustration of the example in Figure 1. Here we assume an ideal model fully trained for RoT.

---

**Step 1**

The context is initialized with the question $Q$.

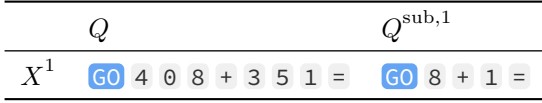

---

**Step 2**

The model generates the first subquestion $8 + 1$.

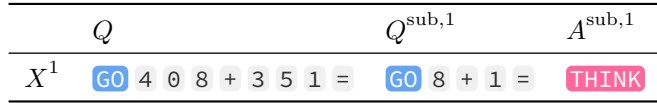

---

**Step 3**

Instead of immediately producing the answer, the model outputs the THINK token.

---

**Step 4**

The THINK token triggers the creation of a new context. The new context is initialized with the subproblem starting from the last GO of $X^1$, i.e., $8 + 1$.

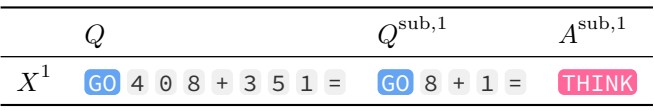

**Step 5**

Since the subproblem is a base case, the model outputs the answer 9 immediately.

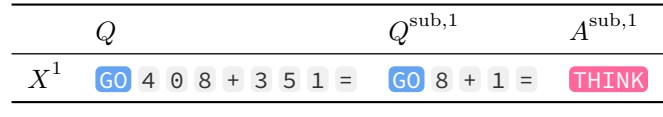

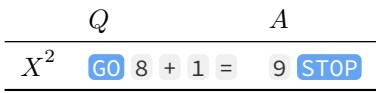

**Step 6**

The answer is returned and replaces the THINK token.

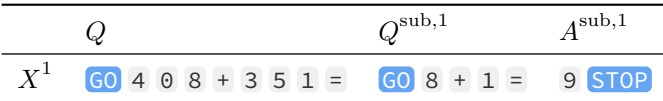

**Step 7**

The model generates the next subproblem, which is to add the remaining digits. Then, it produces THINK to find its answer.

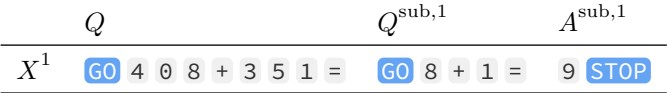

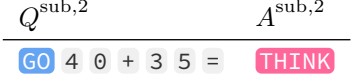

**Step 8**

The THINK token creates a new context $X^3$ for solving $40 + 35$.

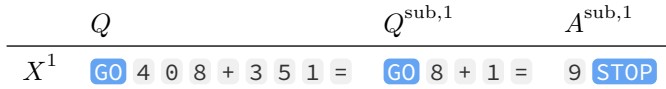

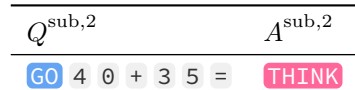

## Step 9

Since $40 + 35$ is not a base case, the model recursively produces more subproblems. In this case, the first subproblem is to add the last digits, i.e., $0$ and $5$. Then it outputs the THINK token to solve the subproblem.

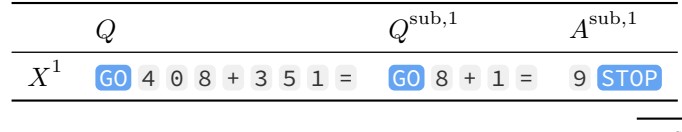

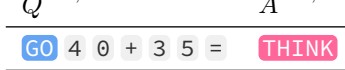

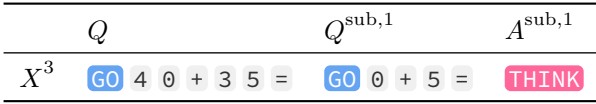

## Step 10

The new context $X^4$ is created to solve $0 + 5$.

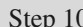

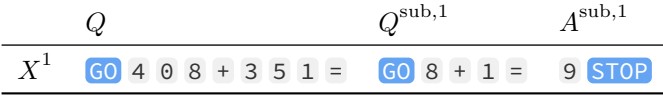

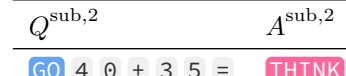

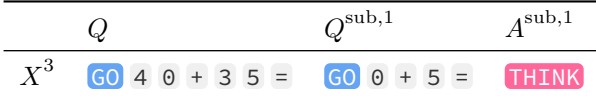

**Step 11**

The answer is returned to $X^3$ and replaces the THINK token.

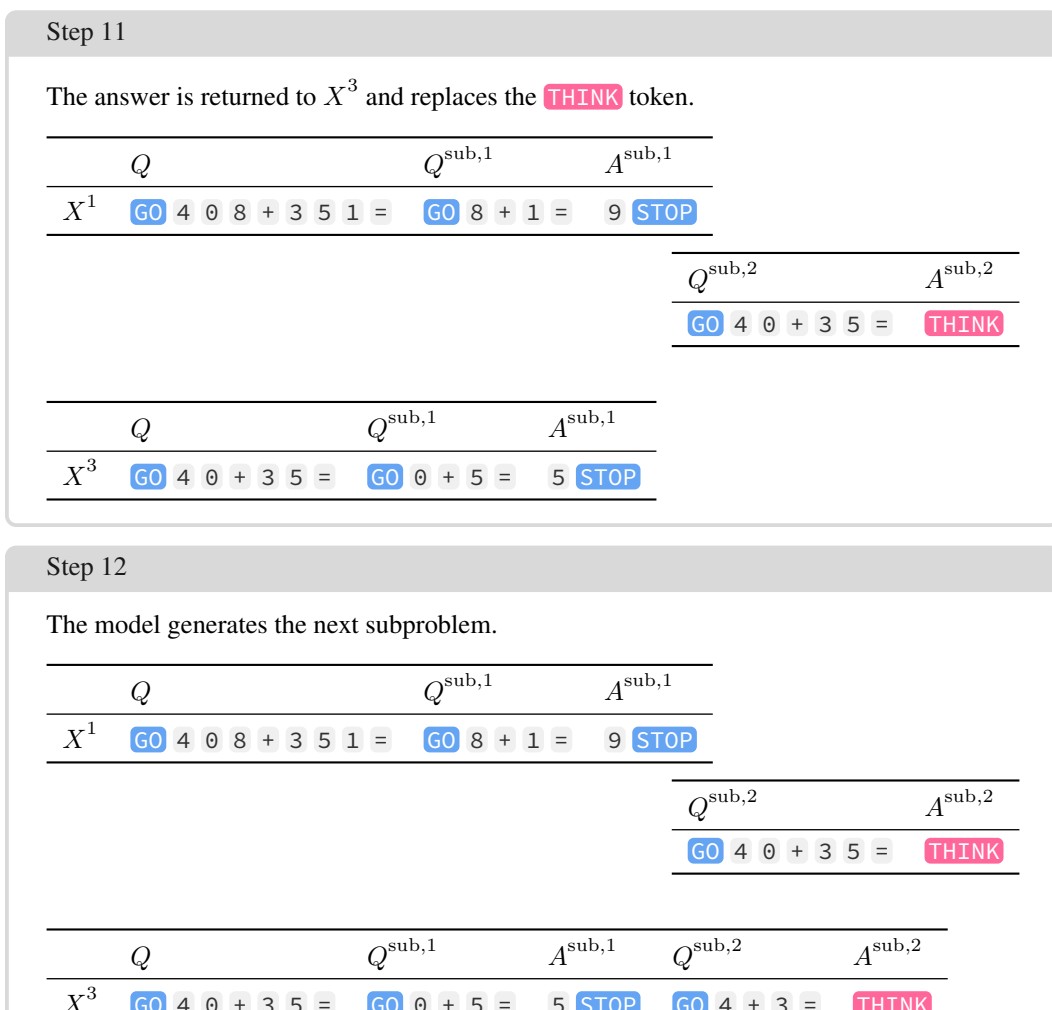

**Step 12**

The model generates the next subproblem.

Step 13

$X^5$ created to solve the subproblem $4 + 3$. Since this is a base case, the model produces the answer directly.

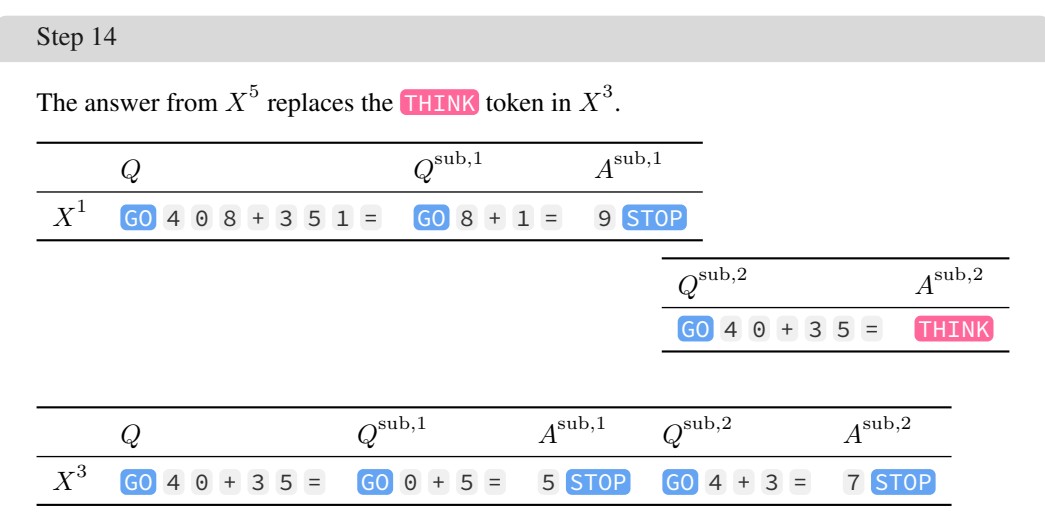

Step 14

The answer from $X^5$ replaces the THINK token in $X^3$.

**Step 15**

Since all subproblems are solved in $X^3$, the answer 75 is generated and returned to $X^1$.

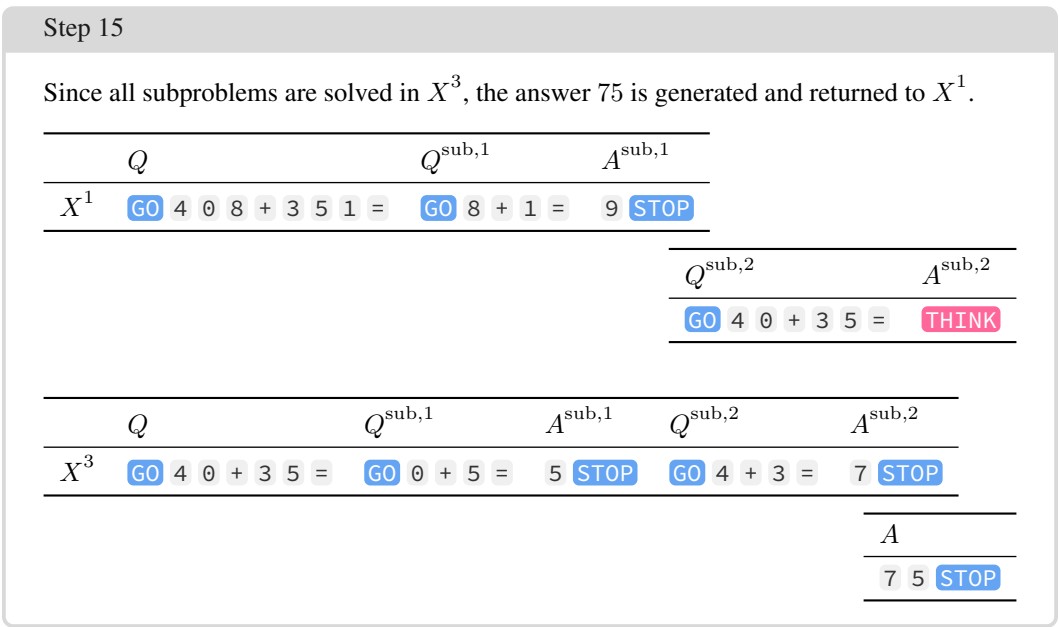

**Step 16**

The answer of $X^3$ replaces the THINK token in $X^1$.

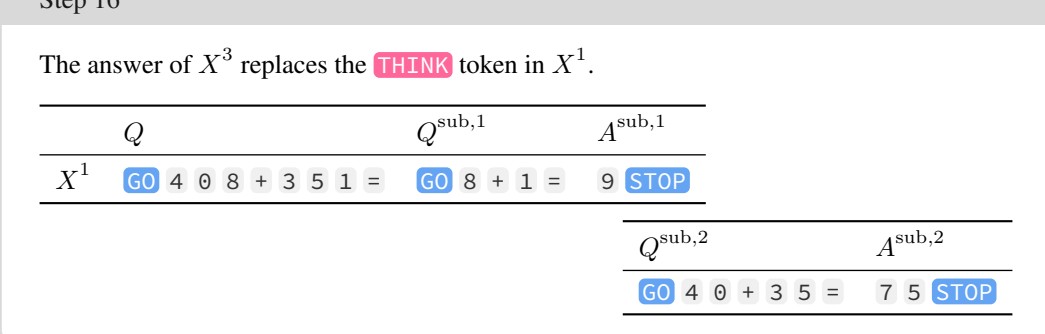

**Step 17**

Since the subproblems in $X^1$ are all solved, the model produces the final answer.

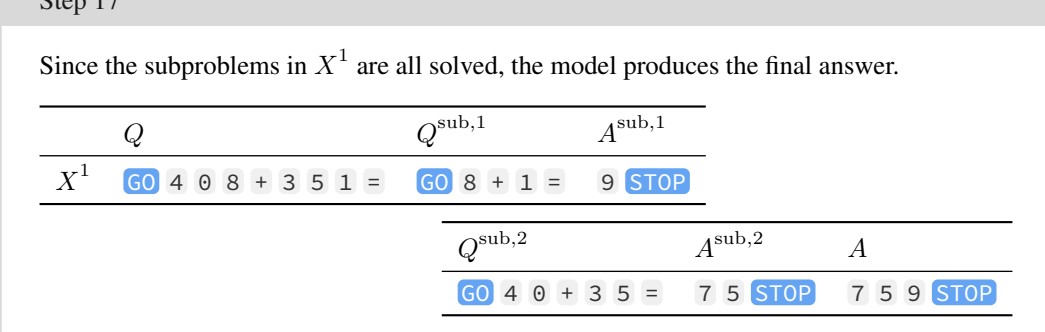

## B  EXAMPLES OF CoT TRAINING DATA

If we solve the example of 408+351 in figure 1 with RoT, the following five contexts are produced.

- $X^1$: GO 4 0 8 + 3 5 1 = GO 8 + 1 = 9 STOP GO 4 0 + 3 5 = 7 5 STOP 7 5 9 STOP
- $X^2$: GO 8 + 1 = 9 STOP
- $X^3$: GO 4 0 + 3 5 = GO 0 + 5 = 5 STOP GO 4 + 3 = 7 STOP 7 5 STOP
- $X^4$: GO 0 + 5 = 5 STOP

- $X^5$: `GO 4 + 3 = 7 STOP`

The CoT context of the same problem is:

- $X^{\mathrm{CoT}}$: `GO 4 0 8 + 3 5 1 = GO 8 + 1 = 9 STOP GO 4 0 + 3 5 = GO 0 + 5 STOP GO 4 + 3 STOP 7 5 STOP 7 5 9 STOP`

In a slightly more complicated example of $34 \times 5$, the RoT contexts are as follows:

- $X^1$: `GO 3 4 * 5 = GO 4 * 5 = 2 0 STOP GO 3 * 5 = 1 5 STOP TAIL 1 5 0 + 2 0 = THINK`
- $X^2$: `GO 4 * 5 = 2 0 STOP`
- $X^3$: `GO 3 * 5 = 1 5 STOP`
- $X^4$: `GO 1 5 0 + 2 0 = GO 0 + 0 = 0 STOP GO 1 5 + 2 = 1 7 STOP 1 7 0 STOP`
- $X^5$: `GO 0 + 0 = 0 STOP`
- $X^6$: `GO 1 5 + 2 = GO 5 + 2 = 7 STOP 1 7 STOP`
- $X^7$: `GO 5 + 2 = 7 STOP`

The corresponding CoT context is:

- $X^{\mathrm{CoT}}$: `GO 3 4 * 5 = GO 4 * 5 = 2 0 STOP GO 3 * 5 = 1 5 STOP TAIL 1 5 0 + 2 0 = GO 0 + 0 = 0 STOP GO 1 5 + 2 = GO 5 + 2 = 7 STOP 1 7 STOP 1 7 0 STOP`

Notice that the CoT context consists of all the corresponding RoT contexts as its subsequences. The number of tokens to generate is identical to that of RoT, if we do not count the `THINK` tokens. Even in these simple examples, however, the context size of CoT is far longer than that of RoT. For much more complex problems, such as 8-digit multiplication or 0-1 Knapsack, the CoT context size can be orders of magnitude larger than RoT. See Appendix I for more details on the distribution of context sizes.

## C  PROBLEM SPECIFICATIONS

### C.1  THE ARITHMETIC PROBLEMS

For arithmetic tasks, we test addition, subtraction, multiplication, and division on non-negative integers. For subtraction, we add a constraint that the first operand is not less than the second one, in order to enforce non-negative answers. For division, we let the output include both a quotient and a remainder, separated by a special token `R`, e.g., `GO 7 ÷ 3 = 2 R 1 STOP`.

As briefly mentioned in §4.1, naively sampling the operands from a uniform distribution makes the operands extremely biased towards large numbers. For example, the probability of sampling a 2-digit number from the 6-digit space is less than 0.01%. Thus, we define a variation of the log-uniform distribution (often called the reciprocal distribution) to sample the operands. As a result, we obtain roughly the same proportion of operands for each number of digits.

The probability density of a log-uniform distribution is proportional to the reciprocal of the value. By definition, zero is not the support of a log-uniform distribution, and samples are overly concentrated to the first few values in the sampling range. Therefore, we slightly extend the log-uniform distribution by introducing an offset parameter $\delta$. To sample an integer in range $[\alpha, \beta)$ with offset $\delta$, we first uniformly sample a real number $r$ in range $[\log(\alpha + \delta), \log(\beta + \delta)]$. Then, $r$ is transformed to $\lfloor \exp(r) - \delta \rfloor$. We denote the extended log-uniform distribution $U_{\log}(\alpha, \beta, \delta)$. As $\delta$ gets larger, the samples are more dispersed to larger numbers. In the experiments, we set $\delta = 3$.

Additionally, we introduce several other sampling details for division problems. Assume that we independently sample two numbers $a$ and $b$ for the dividend and the divisor. In about half of the

| Addition | Subtraction | Multiplication | Division |
|---|---|---|---|
| $1330 + 121163$ | $376776 - 35241$ | $9466 \times 176175$ | $620261 \div 155034$ |
| $114780 + 4356$ | $10638 - 100$ | $179 \times 516$ | $111730 \div 1176$ |
| $638 + 2$ | $109033 - 52649$ | $5509 \times 133$ | $28268 \div 1$ |
| $35 + 77$ | $85137 - 3098$ | $6783 \times 2$ | $588137 \div 25571$ |
| $114261 + 354$ | $22355 - 2824$ | $6 \times 80285$ | $180330 \div 739$ |
| $3 + 13792$ | $7 - 1$ | $37275 \times 19258$ | $879975 \div 97772$ |
| $10151 + 7$ | $652781 - 78853$ | $168484 \times 154$ | $111461 \div 905026$ |
| $22 + 1399$ | $64914 - 3114$ | $3331 \times 40$ | $42338 \div 14003$ |
| $363356 + 450475$ | $13041 - 1422$ | $349 \times 158$ | $108 \div 384103$ |
| $73 + 11$ | $28293 - 4540$ | $17988 \times 262130$ | $60002 \div 7479$ |
| $179895 + 4128$ | $11553 - 3576$ | $8140 \times 1670$ | $131467 \div 131290$ |
| $3 + 10$ | $656291 - 2795$ | $51 \times 5$ | $890679 \div 62$ |
| $1 + 141972$ | $93 - 42$ | $16497 \times 158$ | $228 \div 131108$ |
| $57612 + 18403$ | $55972 - 1782$ | $74 \times 10$ | $892 \div 124$ |
| $9 + 1621$ | $84587 - 51$ | $216 \times 13414$ | $15 \div 964156$ |
| $3370 + 381$ | $273269 - 5867$ | $621 \times 2$ | $369044 \div 28364$ |
| $678 + 8854$ | $274405 - 14$ | $2 \times 5951$ | $457 \div 46$ |
| $422 + 10348$ | $51926 - 9$ | $189486 \times 13080$ | $14687 \div 730$ |
| $118 + 582$ | $4272 - 229$ | $552792 \times 763$ | $200361 \div 1049$ |
| $1343 + 408534$ | $223267 - 377$ | $77 \times 3$ | $19715 \div 965179$ |
| $24 + 9251$ | $14857 - 1994$ | $179090 \times 469029$ | $98 \div 7$ |
| $315 + 652424$ | $914771 - 836$ | $1037 \times 258$ | $406 \div 9$ |
| $355 + 4434$ | $3035 - 2963$ | $8 \times 769974$ | $47345 \div 122$ |
| $22 + 834928$ | $30 - 12$ | $47765 \times 7254$ | $391613 \div 1631$ |
| $3028 + 357$ | $149 - 4$ | $5608 \times 18164$ | $892642 \div 3898$ |
| $777 + 1355$ | $89057 - 6$ | $21437 \times 12$ | $241554 \div 1901$ |
| $154874 + 81059$ | $296410 - 9$ | $15007 \times 15$ | $116475 \div 12908$ |
| $64936 + 216852$ | $45 - 3$ | $539860 \times 427$ | $488317 \div 197443$ |
| $3 + 340939$ | $78906 - 3$ | $3583 \times 9754$ | $7519 \div 325$ |
| $3 + 984775$ | $56560 - 29960$ | $13 \times 66$ | $3560 \div 847611$ |
| $50581 + 1183$ | $98 - 6$ | $266394 \times 185$ | $9711 \div 1385$ |
| $415 + 943$ | $16551 - 920$ | $3988 \times 12$ | $44540 \div 103$ |
| $110 + 49$ | $25606 - 194$ | $5514 \times 57$ | $19721 \div 58$ |
| $15 + 17058$ | $45 - 37$ | $5 \times 1712$ | $59544 \div 24$ |
| $36278 + 100$ | $129443 - 70196$ | $17 \times 430178$ | $333057 \div 333057$ |
| $6 + 23516$ | $221 - 54$ | $227 \times 127$ | $25719 \div 5142$ |
| $1462 + 848$ | $11010 - 818$ | $20888 \times 54$ | $7544 \div 46$ |
| $1002 + 2773$ | $47759 - 67$ | $96 \times 232801$ | $45 \div 410$ |
| $135 + 178346$ | $10 - 8$ | $175 \times 1050$ | $195659 \div 2047$ |
| $22672 + 162038$ | $1439 - 153$ | $146 \times 166$ | $412572 \div 16$ |

Table 1: 40 randomly selected samples of each type of 6-digit arithmetic problems.

cases, the dividend $a$ would be less than the divisor $b$, so the quotients will be zero for those cases. To ensure a diverse range of quotients, we sample the divisor $b$ from $U_{\log}(1, 10^N, 3)$, the quotient $c$ from $U_{\log}(0, 10^N/b, 3)$, and the remainder $r$ from $U_{\log}(0, b, 3)$. The dividend is calculated from these values: $a = b \times c + r$. This way, we can sample division problems with a diverse range of quotients and remainders.

Table 1 presents 40 problem samples for each 6-digit problem type. Several properties of our sampling scheme can be observed from the table. First, each number ranges over diverse numbers of digits. Second, the division problems are mostly non-trivial, i.e., the quotients are not concentrated at zero.

## C.2 THE ALGORITHMIC PROBLEMS

### C.2.1 LONGEST COMMON SUBSEQUENCE (LCS)

The question of an LCS problem is two number sequences joined by the `LCS` token, and the answer is the corresponding LCS and its length separated by `;`. Here is an example of length-4 LCS problem:

- $Q$: `GO 1 2 3 4 LCS 2 4 6 8 =`
- $A$: `2 4 ; 2 STOP`

For a length-$N$ LCS problem, we sample two sequences of length $N$. Each character of the sequences are randomly sampled from 0-9 with equal probability.

### C.2.2 LONGEST PALINDROMIC SUBSEQUENCE (LPS)

The question of a length-$N$ LPS problem starts with the `LPS`, followed by a sequence of length $N$. Similar to LCS, the answer contains the corresponding LPS and its length separated by `;`. The following is an example of length-8 LPS problem:

- $Q$: `GO LPS 4 1 2 5 3 2 6 1 =`
- $A$: `1 2 3 2 1 ; 5 STOP`

The sequence of an LPS problem is sampled in the same way as done for the LCS problem.

### C.2.3 0-1 KNAPSACK

Each item in a 0-1 Knapsack problem is represented by its value and weight. For instance, `1 2 & 3 4` represents an item with a value of 12 and a weight of 34. The question part of a 0-1 Knapsack problem is a sequence consisting of the `KNAPSACK` token, a list of items separated by `,`, the token `@`, and the capacity of the knapsack. The answer part starts with a list of items to include, then `$`, and finally the total value. The following is an example of a 3-item knapsack problem.

- $Q$: `GO KNAPSACK 5 & 1 2 , 2 5 & 1 5 , 1 9 & 1 8 @ 4 0 =`
- $A$: `2 5 & 1 5 , 1 9 & 1 8 $ 4 4 STOP`

In this example, given a knapsack of capacity 40, the last two are selected with the total value of 44.

For a fixed number of items, we uniformly sample each item's value and weight from the integers of range [1, 99].

### C.2.4 MATRIX CHAIN MULTIPLICATION (MCM)

The cost of multiplying many matrices is very sensitive to the order of multiplication. Matrix chain multiplication is the task of finding the best order with the minimum cost. Here, the cost is defined to be the total number of element multiplications. In the example of three matrices $A$, $B$, and $C$, whose shapes are $4 \times 2$, $2 \times 8$, and $8 \times 3$ respectively, the cost of computing $(AB)C$ is $4 \times 2 \times 8 + 4 \times 8 \times 3 = 160$, while another order $A(BC)$ costs only $2 \times 8 \times 3 + 4 \times 2 \times 3 = 72$. In the question of an MCM problem, the sizes of the matrices are enumerated, and the answer contains the order and the total cost separated by `;`. The example above is represented as the following sequences.

- $Q$: `GO MCM 4 × 2 , 2 × 8 , 8 × 3 =`
- $A$: `4 × 2 , ( 2 × 8 , 8 × 3 ) ; 7 2 STOP`

Given a fixed number of matrices, we sample the sizes of matrices from the range [1, 99].

### C.2.5 SORTING

Although not included in the main text, we test the problem of sorting multi-digit numbers. The results are presented in Appendix J. The problem difficulty is defined by the maximum number of

terms. For a sorting problem of at most $N$ terms, we first uniformly sample the number of terms from $[2, N]$. Then we sample each term from $U_{\log}(0, 1000, 5)$. The following is an example of the sorting problem.

- $Q$: `GO SORT 1 3 9 , 1 6 0 , 4 3 4 , 7 9 6 , 4 1 =`
- $A$: `4 1 , 1 3 9 , 1 6 0 , 4 3 4 , 7 9 6 STOP`

## D  DETAILS OF THE RECURSIVE REASONING PROCEDURES

In this section, we elaborate the procedures to recursively solve the arithmetic problems. Specifically, we present the algorithms to produce the subproblems of a problem. Therefore, for a set of randomly sampled questions, we can generate ground truth contexts using these algorithms. For better understanding, we present the key parts of our Python code, the `thought` methods. For each problem, we create a child class the `Problem` class and implement `thought` static method. The method takes a set of arguments for a problem and returns the list of direct subproblems. Each subproblem is represented by a problem class, problem arguments, and recursion type (whether it is a tail recursion or not). We use named tuple `T` to group these information:

```
1  from collections import namedtuple
2  T = namedtuple('Thought', ['prob_cls', 'args', 'type'], defaults=[''])
```

For instance, `T(Mul, (3, 4))` represents a regular subproblem of $3 \times 4$, and `T(Add, (12, 340), 'tail')` represents a subproblem of $12 + 340$ which should be performed as a tail recursion. Once the `thought` method returns a list of `T`s, we can recursively find more subproblems for each subproblem.

### D.1  ADDITION

The core idea of our recursive procedure for addition is to first add the last digits, and then add the rest. If the sum of the last digits is greater than or equal to 10, we insert another subproblem for adding the carry right after adding the last digits.

```
1   class Add(Problem):
2       @staticmethod
3       def thought(args) -> list[T]:
4           left, right = args
5   
6           # Base cases
7           if left < 10 and right < 10:
8               return []
9   
10          l_last, r_last = left % 10, right % 10
11          thoughts = [T(Add, (l_last, r_last))]
12  
13          l_rest, r_rest = left // 10, right // 10
14          if l_last + r_last >= 10:
15              thoughts.append(T(Add, (l_rest, 1)))
16              l_rest += 1
17  
18          if l_rest > 0 and r_rest > 0:
19              thoughts.append(T(Add, (l_rest, r_rest)))
20  
21          return thoughts
```

Figure 1 in the main draft is an example with no carry, and the following is another example of 27+65 with a carry.

- $X^1$: `GO` 3 1 7 + 6 5 = `GO` 7 + 5 = 1 2 `STOP` `GO` 3 1 + 1 = 3 2 `STOP` `GO` 3 2 + 6 = 3 8 `STOP` 3 8 2 `STOP`

- $X^2$: `GO` 7 + 5 = 1 2 `STOP`
- $X^3$: `GO` 3 1 + 1 = `GO` 1 + 1 = 2 `STOP` 3 2 `STOP`
- $X^4$: `GO` 1 + 1 = 2 `STOP`
- $X^5$: `GO` 3 2 + 6 = `GO` 2 + 6 = 8 `STOP` 3 8 `STOP`
- $X^6$: `GO` 2 + 6 = 8 `STOP`

## D.2 SUBTRACTION

Similar to addition, we first subtract the last digits and solve the rest recursively. When subtracting the last digits $x$ and $y$, we always borrow 10 for $x$ to prevent a negative result. The borrowing of 10 is easy for a sequence model: just put `1` before $x$. Therefore, the base cases of subtraction are when $a \le 19$ and $b \le 9$. If the subtraction result of the last digits is smaller than 10, i.e., the borrow is actually needed, we subtract 1 from the rest of the first operand $m$.

```python
class Sub(Problem):
    @staticmethod
    def thought(args) -> list[T]:
        left, right = args

        # Base cases
        if left <= 19 and right <= 9:
            return []

        l_last = left % 10 + 10
        r_last = right % 10
        thoughts = [T(Sub, (l_last, r_last))]
        l_rest, r_rest = left // 10, right // 10
        if l_last - r_last < 10:
            thoughts.append(T(Sub, (l_rest, 1)))
            l_rest -= 1
        if r_rest > 0:
            thoughts.append(T(Sub, (l_rest, r_rest)))

        return thoughts
```

Here is an example of 432-216:

- $X^1$: `GO` 4 3 2 − 2 1 6 = `GO` 1 2 − 6 = 6 `STOP` `GO` 4 3 − 1 = 4 2 `STOP` `GO` 4 2 − 2 1 = 2 1 `STOP` 2 1 6 `STOP`
- $X^2$: `GO` 1 2 − 6 = 6 `STOP`
- $X^3$: `GO` 4 3 − 1 = `GO` 1 3 − 1 = 1 2 `STOP` 4 2 `STOP`
- $X^4$: `GO` 1 3 − 1 = 1 2 `STOP`
- $X^5$: `GO` 4 2 − 2 1 = `GO` 1 2 − 1 = 1 1 `STOP` `GO` 4 − 2 = 2 `STOP` 2 1 `STOP`
- $X^6$: `GO` 1 2 − 1 = 1 1 `STOP`
- $X^7$: `GO` 4 − 2 = 2 `STOP`

Notice that the final answer and the questions of each subproblem can be easily constructed from previous sequence.

## D.3 MULTIPLICATION

The base cases of multiplication are (i) when either operands are 0 or 1, or (ii) when both operands are less than 10. If one of the operands is 0, then the answer is zero; when one of them is 1, then the answer is just a copy of the other operand. For the cases where both operands are less than 10, we just let the model memorize them, which is similar to an elementary school math curriculum.

There are two types of non-base cases. For the simpler case, where the second operand is less than 10, we first split the first operand into the last digit and the rest. We then multiply each of them with the second operand and combine the results. Otherwise, we split the second operand into the last digit and the rest. The first operand is multiplied to each of them, and the results are summed.

```python
class Mul(Problem):
    @staticmethod
    def thought(args) -> list[T]:
        left, right = args

        # Base cases
        if left <= 1 or right <= 1:
            return []
        if left <= 9 and right <= 9:
            return []

        thoughts = []
        if right < 10:
            thoughts.append(T(Mul, (left % 10, right)))
            thoughts.append(T(Mul, (left // 10, right)))

            a1 = (left % 10) * right
            a2 = (left // 10) * right
            thoughts.append(T(Add, (a2 * 10, a1), 'tail'))
        else:
            a1 = left * (right % 10)
            thoughts.append(T(Mul, (left, right % 10)))

            a2 = left * (right // 10)
            thoughts.append(T(Mul, (left, right // 10)))

            thoughts.append(T(Add, (a2 * 10, a1), 'tail'))
        return thoughts
```

Here are some example contexts of multiplication:

- $X^1$: GO 4 3 * 2 1 = GO 4 3 * 1 = 4 3 STOP GO 4 3 * 2 = 8 6 STOP TAIL 8 6 0 + 4 3 = THINK

- $X^2$: GO 4 3 * 1 = 4 3 STOP

- $X^3$: GO 4 3 * 2 = GO 3 * 2 = 6 STOP GO 4 * 2 = 8 STOP TAIL 8 0 + 6 = THINK

- $X^4$: GO 3 * 2 = 6 STOP

- $X^5$: GO 4 * 2 = 8 STOP

- $X^6$: GO 8 0 + 6 = GO 0 + 6 = 6 STOP 8 6 STOP

- $X^7$: GO 0 + 6 = 6 STOP

- $X^8$: GO 8 6 0 + 4 3 = GO 0 + 3 = 3 STOP GO 8 6 + 4 = 9 0 STOP 9 0 3 STOP

- $X^9$: GO 0 + 3 = 3 STOP

- $X^{10}$: GO 8 6 + 4 = GO 6 + 4 = 1 0 STOP GO 8 + 1 = 9 STOP 9 0 STOP

- $X^{11}$: GO 6 + 4 = 1 0 STOP

- $X^{12}$: GO 8 + 1 = 9 STOP

Notice that we use tail recursion in $X^1$ and $X^3$.

### D.4 COMPARISON

Comparison is used as a subroutine during division. The procedure for comparison consists of three steps:

1. Compare the numbers of digits.
2. If the numbers of digits are the same, compare the most significant digits.
3. If the most significant digits are identical, compare the remaining digits recursively.

We find that the sequence models can perform the first step without an explicit subproblem. Therefore, we only add intermediate steps for the second and the third steps.

```python
1   class Compare(Problem):
2       @staticmethod
3       def thought(args) -> list[T]:
4           left, right = args
5
6           # Base cases
7           if left < 10 and right < 10:
8               return []
9
10          thoughts = []
11          digit_l, digit_r = len(str(left)), len(str(right))
12          if digit_l == digit_r:
13              # Compare first digit
14              l_first, r_first = int(str(left)[0]), int(str(right)[0])
15              thoughts.append(T(Compare, (l_first, r_first)))
16              if l_first == r_first:
17                  # Compare the rest
18                  l_rest = int(str(left)[1:])
19                  r_rest = int(str(right)[1:])
20                  thoughts.append(T(Compare, (l_rest, r_rest)))
21
22          return thoughts
```

The following is an example of comparing 153 and 159.

- $X^1$: GO 1 5 3 VS 1 5 9 = GO 1 VS 1 = EQ STOP GO 5 3 VS 5 9 = LT STOP LT STOP
- $X^2$: GO 1 VS 1 = EQ STOP
- $X^3$: GO 5 3 VS 5 9 = GO 5 VS 5 = EQ STOP GO 3 VS 9 = LT STOP LT STOP
- $X^4$: GO 5 VS 5 = EQ STOP
- $X^5$: GO 3 VS 9 = LT STOP

### D.5 DIVISION

Solving division is the most challenging among the four basic arithmetic operations since the procedure is basically trial and error, searching for the correct quotient. Nonetheless, the following process is a recursive version of the elementary school division.

The base case is when the dividend is less than or equal to the divisor. If the dividend is smaller than the divisor, the quotient is 0, and the remainder is the dividend. If the dividend is equal to the divisor, than the quotient is 1, and the remainder is 0. Both cases can be handled relatively easily by neural sequence models. To determine whether it is one of these cases, we always perform the comparison as the first subproblem.

If it is not a base case, we check whether the dividend is smaller than 10 times the divisor. If the dividend is smaller, we subtract the divisor from the dividend and recursively divide the result with the divisor. The final answer is attained by simply adding 1 to the quotient of the smaller division.

To explain the other case, where the dividend is greater than 10 times the divisor, let us call the dividend $a$ and the divisor $b$. First, we split the $a$ into the last digit $x$ and the remaining digits $m$. Then, we divide $m$ with the divisor $b$, i.e., we are solving one-digit-smaller subproblem first. Since we define the division operation to return both a quotient and a remainder, the quotient $q_1 = m/b$ and the remainder $r_1 = m \mod b$ from the subproblem is added to the context. Next, we *concatenate* the remainder and $x$, which is numerically computing $r \times 10 + x$, and divide it again with $b$. Let the quotient and the remainder of this operation $q_2$ and $r_2$. Then, the quotient of the final answer is $q_1 \times 10 + q_2$, while the remainder is simply $r_2$.

```python
class Div(Problem):
    @staticmethod
    def thought(args) -> list[T]:
        left, right = args
        thoughts = [T(Compare, (left, right))]

        # Base cases
        if left <= right:
            return thoughts

        thoughts.append(T(Compare, (left, right * 10)))
        if left <= right * 10:
            diff = left - right
            thoughts.append(T(Sub, (left, right)))
            thoughts.append(T(Div, (diff, right)))
        else:
            thoughts.append(T(Div, (left // 10, right)))
            left_remainder = (left // 10) % right * 10 + left % 10
            thoughts.append(T(Div, (left_remainder, right)))
        return thoughts
```

The following is an example of $76 \div 29$.

- $X^1$: GO 7 6 ÷ 2 9 = GO 7 6 VS 2 9 = GT STOP GO 7 6 VS 2 9 0 = LT STOP GO 7 6 − 2 9 = 4 7 STOP GO 4 7 ÷ 2 9 = 1 R 1 8 STOP 2 R 1 8 STOP

- $X^2$: GO 7 6 VS 2 9 = GO 7 VS 2 = GT STOP GT STOP

- $X^3$: GO 7 VS 2 = GT STOP

- $X^4$: GO 7 6 VS 2 9 0 = LT STOP

- $X^5$: GO 7 6 − 2 9 = GO 1 6 − 9 = 7 STOP GO 7 − 1 = 6 STOP GO 6 − 2 = 4 STOP 4 7 STOP

- ...

- $X^9$: GO 4 7 ÷ 2 9 = GO 4 7 VS 2 9 = GT STOP GO 4 7 VS 2 9 0 = LT STOP GO 4 7 − 2 9 = 1 8 STOP GO 1 8 ÷ 2 9 = 0 R 1 8 STOP 1 R 1 8 STOP

- $X^{10}$: GO 4 7 VS 2 9 = GO 4 VS 2 = GT STOP GT STOP

- $X^{11}$: GO 4 VS 2 = GT STOP

- $X^{12}$: GO 4 7 VS 2 9 0 = LT STOP

- $X^{13}$: GO 4 7 − 2 9 = GO 1 7 − 9 = 8 STOP GO 4 − 1 = 3 STOP GO 3 − 2 = 1 STOP 1 8 STOP

- ...

- $X^{17}$: GO 1 8 ÷ 2 9 = GO 1 8 VS 2 9 = LT STOP 0 R 1 8 STOP

- $X^{18}$: GO 1 8 VS 2 9 = GO 1 VS 2 = LT STOP LT STOP

- ...

### D.6 LONGEST COMMON SUBSEQUENCE (LCS)

Given sequences $A$ and $B$, the algorithm starts by comparing the last characters of the two sequences. If the last two characters are the same, we find LCS of the subsequences without the last characters, i.e., LCS of $A_{:-1}$ and $B_{:-1}$. Otherwise, we compute the LCSs of the cases where the last character of either side is removed, and return the better one. In the following code, `LCS._answer` is the subroutine that finds the LCS of two sequences. `Equal` returns `TRUE` if the two arguments are the same, or `FALSE` otherwise.

```
1   class LCS(Problem):
2       @staticmethod
3       def thought(args) -> list[T]:
4           l, r = args
5           if len(l) == 0 or len(r) == 0:
6               return []
7
8           thoughts = [T(Equal, (l[-1], r[-1]))]
9           if l[-1] == r[-1]:
10              thoughts.append(T(LCS, (l[:-1], r[:-1])))
11              return thoughts
12
13          lcs1_args = (l[:-1], r)
14          lcs2_args = (l, r[:-1])
15          lcs1 = LCS._answer(lcs1_args)
16          lcs2 = LCS._answer(lcs2_args)
17          thoughts.extend([
18              T(LCS, lcs1_args),
19              T(LCS, lcs2_args),
20              T(Compare, (len(lcs1), len(lcs2)))
21          ])
22          return thoughts
```

The following is an example of finding the LCS of 123 and 234.

- $X^1$: GO 1 2 3 LCS 2 3 4 = GO EQUAL 3 , 4 = FALSE STOP GO 1 2 LCS 2 3 4 = 2 ; 1 STOP GO 1 2 3 LCS 2 3 = 2 3 ; 2 STOP GO 1 VS 2 = LT STOP 2 3 ; 2 STOP

- $X^2$: GO EQUAL 3 , 4 = FALSE STOP

- $X^3$: GO 1 2 LCS 2 3 4 = GO EQUAL 2 , 4 = FALSE STOP GO 1 LCS 2 3 4 = ; 0 STOP GO 1 2 LCS 2 3 = 2 ; 1 STOP GO 0 VS 1 = LT STOP 2 ; 1 STOP

- ...

- $X^{21}$: GO 1 2 3 LCS 2 3 = GO EQUAL 3 , 3 = TRUE STOP GO 1 2 LCS 2 = 2 ; 1 STOP 2 3 ; 2 STOP

- ...

- $X^{23}$: GO 1 VS 2 = LT STOP

### D.7 LONGEST PALINDROMIC SUBSEQUENCE (LPS)

The overall algorithm for LPS is similar to LCS. The base cases are when the sequence length is less then 3. If it is not a base case, we first check if the characters at both ends of the sequence are the same. If they are the same, we find the LPS of the subsequence excluding them. Otherwise, we compare the cases where one of the end characters are excluded.

```
1   class LPS(Problem):
2       @staticmethod
3       def thought(args) -> list[T]:
4           # Base cases
5           if len(args) == 1:
6               return []
```

```
 7            elif len(args) == 2:
 8                return [T(Equal, args)]
 9
10            thoughts = [T(Equal, (args[0], args[1]))]
11            if args[0] == args[-1]:
12                sub_lps = LPS._answer(args[1:-1])
13                thoughts.extend([
14                    T(LPS, args[1:-1]),
15                    T(Add, (len(sub_lps), 2))
16                ])
17            else:
18                lps1_args = args[:-1]
19                lps2_args = args[1:]
20                lps1 = LPS._answer(lps1_args)
21                lps2 = LPS._answer(lps2_args)
22                thoughts.extend([
23                    T(LPS, lps1_args),
24                    T(LPS, lps2_args),
25                    T(Compare, (len(lps1), len(lps2)))
26                ])
27            return thoughts
```

The following is an example of LPS.

- $X^1$: GO LPS 1 2 3 2 = GO EQUAL 1 , 2 = FALSE STOP GO LPS 1 2 3 = 1 ;
  1 STOP GO LPS 2 3 2 = 2 3 2 ; 3 STOP GO 1 VS 3 = LT STOP 2 3 2 ; 3
  STOP

- $X^2$: GO EQUAL 1 , 2 = FALSE STOP

- $X^3$: GO LPS 1 2 3 = GO EQUAL 1 , 3 = FALSE STOP GO LPS 1 2 = 1 ; 1
  STOP GO LPS 2 3 = 2 ; 1 STOP GO 1 VS 1 = EQ STOP 1 ; 1 STOP

- ...

- $X^{10}$: GO LPS 2 3 2 = GO EQUAL 2 , 2 = TRUE STOP GO LPS 3 = 3 ; 1 STOP
  GO 1 + 2 = 3 STOP 2 3 2 ; 3 STOP

- ...

- $X^{14}$: GO 1 VS 3 = LT STOP

### D.8    0-1 KNAPSACK

The base cases are when there is only one item. In this case, we simply compare the item's weight and the knapsack's capacity, to determine whether the item should be included. If it is a non-base case, we compare two possibilities: (i) include the first item, or (ii) exclude the first item. We recursively compute the subproblems and find the case with the best value.

```
 1  class LPS(Problem):
 2      @staticmethod
 3      def thought(args) -> list[T]:
 4          items, capacity = args
 5          value, weight = items[0]
 6
 7          # Base case
 8          if len(items) == 1:
 9              return [T(Compare, (weight, capacity))]
10
11          # When excluding the current item
12          items_max, value_max = Knapsack._answer((items[1:], capacity))
13          thoughts = [
14              T(Knapsack, (items[1:], capacity)),
15              T(Compare, (weight, capacity)),
16          ]
17
```

```
18              # When including the current item
19              if weight <= capacity:
20                  items_sub, value_sub = Knapsack._answer(
21                      (items[1:], capacity - weight))
22                  value_incl = value_sub + value
23                  thoughts.extend([
24                      T(Sub, (capacity, weight)),
25                      T(Knapsack, (items[1:], capacity - weight)),
26                      T(Add, (value_sub, value)),
27                      T(Compare, (value_incl, value_max)),
28                  ])
29
30          return thoughts
```

The following is an example of 0-1 knapsack problem with three items and a knapsack capacity of 10.

- $X^1$: GO KNAPSACK 3 & 9 , 4 & 2 , 9 & 5 @ 1 0 = GO KNAPSACK 4 & 2 , 9 & 5 @ 1 0 = 4 & 2 , 9 & 5 $ 1 3 STOP GO 9 VS 1 0 = LT STOP GO 1 0 - 9 = 1 STOP GO KNAPSACK 4 & 2 , 9 & 5 @ 1 = $ 0 STOP GO 0 + 3 = 3 STOP GO 3 VS 1 3 = LT STOP 4 & 2 , 9 & 5 $ 1 3 STOP

- $X^2$: GO KNAPSACK 4 & 2 , 9 & 5 @ 1 0 = GO KNAPSACK 9 & 5 @ 1 0 = 9 & 5 $ 9 STOP GO 2 VS 1 0 = LT STOP GO 1 0 - 2 = 8 STOP GO KNAPSACK 9 & 5 @ 8 = 9 & 5 $ 9 STOP GO 9 + 4 = 1 3 STOP GO 1 3 VS 9 = GT STOP 4 & 2 , 9 & 5 $ 1 3 STOP

- ...

- $X^{11}$: GO 9 VS 1 0 = LT STOP

- $X^{12}$: GO 1 0 - 9 = 1 STOP

- $X^{13}$: GO KNAPSACK 4 & 2 , 9 & 5 @ 1 = GO KNAPSACK 9 & 5 @ 1 = $ 0 STOP GO 2 VS 1 = GT STOP $ 0 STOP

- ...

- $X^{17}$: GO 0 + 3 = 3 STOP

- $X^{18}$: GO 3 VS 1 3 = LT STOP

### D.9 TERNARY ADDITION AND MULTIPLICATION

Ternary addition and multiplication arises as a subproblem while solving MCM, which will be explained in the next section. They are simple extensions of addition and multiplication to three integers.

```
1   class TernaryAdd(Problem):
2       @staticmethod
3       def thought(args) -> list[T]:
4           a1, a2, a3 = args
5           return [
6               T(Add, (a1, a2)),
7               T(Add, (a1 + a2, a3), 'tail')
8           ]
9
10
11  class TernaryMul(Problem):
12      @staticmethod
13      def thought(args) -> list[T]:
14          a1, a2, a3 = args
15          return [
16              T(Mul, (a1, a2)),
17              T(Mul, (a1 * a2, a3), 'tail')
18          ]
```

## D.10 MATRIX CHAIN MULTIPLICATION (MCM)

Given $N$ matrices, the $N - 1$ subproblems are defined for each possible binary split. For the multiplication of four matrices $ABCD$, there are three possible binary splits: $A(BCD)$, $(AB)(CD)$, and $(ABC)D$. For each binary split, the total cost is the sum of (i) the minimum cost of computing the first group, (ii) the minimum cost of computing the second group, and (iii) the cost of multiplying the two matrices resulting from each group. Once we get the total costs of each binary split, we return choose the best split with the minimum cost. The following code implements this procedure.

```python
class MCM(Problem):
    @staticmethod
    def thought(args) -> list[T]:
        mats, min_order, min_cost = args

        # Base cases
        if len(mats) == 1:
            return []

        if min_order is None:
            # Top-level problem
            l_mats, r_mats = mats[:1], mats[1:]
        else:
            # Middle of recursion
            l_mats, r_mats = mats

        l_args = (l_mats, None, None)
        r_args = (r_mats, None, None)
        l_order, l_cost = MCM._answer(l_args)
        r_order, r_cost = MCM._answer(r_args)
        agg_cost = l_mats[0][0] * r_mats[0][0] * r_mats[-1][1]
        thoughts = [
            T(MCM, l_args),
            T(MCM, r_args),
            T(TernaryMul, (l_mats[0][0], r_mats[0][0], r_mats[-1][1])),
            T(TernaryAdd, (l_cost, r_cost, agg_cost)),
        ]

        cost = l_cost + r_cost + agg_cost
        if min_cost is not None:
            thoughts.append(T(Compare, (cost, min_cost)))
        if min_cost is None or cost < min_cost:
            min_cost = cost
            min_order = l_order, r_order

        if len(r_mats) > 1:
            new_l_mats = l_mats + (r_mats[0],)
            new_r_mats = r_mats[1:]
            thoughts.append(
                T(MCM, ((new_l_mats, new_r_mats), min_order, min_cost), 'tail'))

        return thoughts
```

The following is an example of three-matrix MCM.

- $X^1$: `GO` MCM 3 × 9 , 9 × 4 , 4 × 5 = `GO` MCM 3 × 9 = 3 × 9 ; 0 `STOP` `GO` MCM 9 × 4 , 4 × 5 = 9 × 4 , 4 × 5 ; 1 8 0 `STOP` `GO` 3 ⋆ 9 ⋆ 5 = 1 3 5 `STOP` `GO` 0 + 1 8 0 + 1 3 5 = 3 1 5 `STOP` `TAIL` MCM 3 × 9 , 9 × 4 | 4 × 5 ACC 3 × 9 , ( 9 × 4 , 4 × 5 ) ; 3 1 5 = `THINK`

- ...

- $X^{32}$: `GO` MCM 3 × 9 , 9 × 4 | 4 × 5 ACC 3 × 9 , ( 9 × 4 , 4 × 5 ) ; 3 1 5 = `GO` MCM 3 × 9 , 9 × 4 = 3 × 9 , 9 × 4 ; 1 0 8 `STOP` `GO` MCM 4 × 5 = 4 × 5 ; 0 `STOP` `GO` 3 ⋆ 4 ⋆ 5 = 6 0 `STOP` `GO` 1 0 8 + 0 + 6 0 = 1 6 8

`STOP` `GO` `1` `6` `8` `VS` `3` `1` `5` `=` `LT` `STOP` `(` `3` `×` `9` `,` `9` `×` `4` `)` `,` `4` `×` `5` `;` `1` `6` `8` `STOP`

- ...

### D.11   SORTING

Among several sorting algorithms, we choose merge sort for our experiments with CoT and RoT. Note that WT is not relevant to the sorting algorithm since it produces the answer directly. The merge sort algorithm is simple: (i) split the given sequence to two equally sized subsequences, (ii) sort each subsequence, and (iii) merge the two sorted sequences. Since the final merge operation is quite complicated, we define the merge as a problem type.

```python
class Merge(Problem):
    @staticmethod
    def thought(args) -> list[T]:
        l, r = args
        if len(l) == 0 or len(r) == 0:
            return []

        thoughts = [T(Compare, (l[0], r[0]))]
        if l[0] < r[0] and len(l) > 1:
            thoughts.append(T(Merge, (l[1:], r)))
        elif l[0] >= r[0] and len(r) > 1:
            thoughts.append(T(Merge, (l, r[1:])))
        return thoughts

class MergeSort(Problem):
    @staticmethod
    def thought(args) -> list[T]:
        if len(args) < 2:
            return []

        l_len = (len(args) + 1) // 2
        l = args[:l_len]
        r = args[l_len:]
        return [
            T(MergeSort, l),
            T(MergeSort, r),
            T(Merge, (tuple(sorted(l)), tuple(sorted(r))), 'tail')
        ]
```

## E   FINE-TUNING GPT-3 FOR RECURSION OF THOUGHT

Using the OpenAI API, we fine-tune GPT-3 for Recursion of Thought. The goal is to learn 16-digit addition, 16-digit subtraction, 8-digit multiplication, and 8-digit division simultaneously. GPT-3's fine-tuning API takes a dataset where each example is a prompt-completion pair in plain text. It is converted to tokens by a special tokenizer for GPT, which we cannot control. This API is not directly compatible with RoT due to several reasons.

- There is no special tokens such as `GO`, `THINK`, and `STOP`.
- The input and target sequences have to be the same. However, they are different in RoT due to the `THINK` token. Once `THINK` is produced, the RoT framework triggers the recursion process to find the subproblem's answer and replace the `THINK` token with it. Therefore, the `THINK` token appears in the target sequences, but never in the input sequences.

Moreover, the way that GPT-3 tokenizes numbers hinders the learning of arithmetic reasoning rules. GPT-3 tokenizes a multi-digit number into a set of two-digit or three-digit numbers. For example, the text `1234567` is converted to the sequence of tokens `123` `45` `67`. Under this tokenization scheme, the relationship between the numbers become obscured. As an example, the tokens `7`, `17`,

`27`, ..., `997` all have 7 as their last digit. Since there is no direct way for a model to know that they share the same digit, it is crucial to use each digit as a token. We believe that OpenAI needs to correct this tokenization of GPT-3 for numbers.

Luckily, we can mimic the RoT procedures with the API by using several tricks. First, we replace the special tokens with plain lower-case words, e.g., `GO` → `go` and `STOP` → `stop`, which are included in the vocabulary of GPT-3. Second, we add a space before each token to make sure that the GPT tokenizer separates each token. We also add space before each digit to prevent the tokenizer grouping a number into 2-to-3-digit tokens. Finally, to simulate the behavior of the `THINK` and `STOP` tokens, we derive multiple examples from each context, one for each `THINK` or `STOP` output.

As an example, context $X^3$ in Figure 1 is converted to the following JSON lines for GPT-3 as follows:

| $X^3$ | GO 4 0 + 3 5 = | GO 0 + 5 = 5 STOP | GO 4 + 3 = 7 STOP | 7 5 STOP |
|---|---|---|---|---|
| $Y^3$ | PAD ×7 | GO 0 + 5 = THINK PAD | GO 4 + 3 = THINK PAD | 7 5 STOP |

$$\Downarrow$$

```
1  {"prompt": "_go_4_0_+_3_5_=", "completion": "_go_0_+_5_=_think"}
2  {"prompt": "_go_4_0_+_3_5_=_go_0_+_5_=_5_stop", "completion": "_go_4_+_3_=_think"}
3  {"prompt": "_go_4_0_+_3_5_=_go_0_+_5_=_5_stop_go_4_+_3_=_7_stop", "completion": "_
      7_5_stop"}
```

In the case of Without Thought (WT), each problem is simply converted into a single example:

| $X$ | GO 4 0 + 3 5 = | 7 5 STOP |
|---|---|---|
| $Y$ | PAD ×7 | 7 5 STOP |

$$\Downarrow$$

```
1  {"prompt": "_go_4_0_+_3_5_=", "completion": "_7_5_stop"}
```

In both cases of RoT and WT, we fine-tune GPT-3 for 10K steps with a batch size of 256. Among the several variants of GPT-3, we use Ada which is offered at the lowest cost. Note that RoT produces multiple contexts for each problem, and each RoT context is converted to multiple training examples. For this reason, the GPT-3 fine-tuned for RoT encounters much fewer problems during training, although the number of training steps are the same.

## F  TRAINING DETAILS OF THE TINY MODELS

In all experiments, we use a batch size of 256 and Adam optimizer Kingma & Ba (2015) with a learning rate of 0.001, i.e., the default learning rate in PyTorch. We train the Transformers for 500K steps and and decay the learning rate by half every 50K steps. Since the LSTMs converge slower than the Transformers, we train them for 800K steps and decay the learning rate by half every 100K steps. At every 20K steps, we evaluate the model on a test set of 30K problems, and if a model reaches a perfect accuracy of 1.0, we do not train the model further. The models can be trained on a single GPU with 12GB memory.

## G  EFFICIENT EVALUATION OF ROT

At the problem scales that RoT is tested, solving a single problem can require hundreds of thousands of tokens. Therefore, we have to develop an efficient evaluation algorithm. Since the models are

deterministic, we do not need to evaluate the same context multiple times. Therefore, given a test set, we first find all unique contexts that should appear while solving the test set. Note that there would be a lot of duplicate contexts, especially for the base cases. These unique contexts can be evaluated in parallel. We evaluate each context in a similar manner to the training. For each context, we construct the target sequence, as we described in §3.2, and check if the model outputs the target sequence correctly, given the GT context. In case of GPT-3, we convert the contexts and target sequences as described in Appendix E to feed into the API. Once every context is individually evaluated, the next step is to aggregate the results. In this aggregation stage, a (sub)problem is marked correct only if its subproblems are all marked correct. This way, we efficiently evaluate RoT with minimal cost.

## H    FAILURE CASE ANALYSIS OF GPT-3 WITH ROT

In this section we provide some failure cases of RoT in the GPT-3 experiments for further insights. Although the model achieves an accuracy of over 99% in almost every case, there are several problems where it makes mistakes. Nonetheless, only a few characters are wrong in each context.

The most common type of error is repeating the same character for a wrong number of times. The following is the examples of this error type.

| Prompt | `go 1 6 4 3 4 vs 1 0 1 1 6 = go 1 vs 1 = eq stop` |
|---|---|
| Completion | `go 6 4 3 4 vs 1 1 6 = think` |
| Model Output | `go 6 4 3 4 vs 1 1 1 6 = think` |

| Prompt | `go 4 8 0 9 4 4 9 5 1 8 2 5 5 9 1 * 4 4 3 0 2 5 3 6 0 1 7`
`= go 4 8 0 9 4 4 9 5 1 8 2 5 5 9 1 * 7 = 3 3 6 6 6 1 4 6`
`6 2 7 7 9 1 3 7 stop go 4 8 0 9 4 4 9 5 1 8 2 5 5 9 1 * 4`
`4 3 0 2 5 3 6 0 1 = 2 1 3 0 7 0 8 1 0 4 7 0 8 0 9 6 0 5 1`
`7 0 3 1 9 1 stop` |
|---|---|
| Completion | `tail 2 1 3 0 7 0 8 1 0 4 7 0 8 0 9 6 0 5 1 7 0 3 1 9 1 0`
`+ 3 3 6 6 6 1 4 6 6 2 7 7 9 1 3 7 = think` |
| Model Output | `tail 2 1 3 0 7 0 8 1 0 4 7 0 8 0 9 6 0 5 1 7 0 3 1 9 1 0`
`+ 3 3 6 6 6 6 1 4 6 6 2 7 7 9 1 3 7 = think` |

| Prompt | `go 1 0 5 4 6 2 0 * 1 5 = go 1 0 5 4 6 2 0 * 5 = 5 2 7 3`
`1 0 0 stop go 1 0 5 4 6 2 0 * 1 = 1 0 5 4 6 2 0 stop` |
|---|---|
| Completion | `tail 1 0 5 4 6 2 0 0 + 5 2 7 3 1 0 0 = think` |
| Model Output | `tail 1 0 5 4 6 2 0 0 0 + 5 2 7 3 1 0 0 = think` |

Another common case is incorrectly counting the number of digits while comparing two numbers. When comparing numbers following our recursive procedure, the model has to immediately return the answer if the number of digits are different. In the following example, the first number has seven digits, and the second one has six digits.

| Prompt | `go 1 0 6 9 0 3 4 vs 8 9 1 5 0 6 =` |
|---|---|
| Completion | `gt stop` |
| Model Output | `go 1 vs 8` |

Although the model should directly output the answer, i.e., `gt stop`, it decides that the number of digits are the same and moves on to the next step, which is to compare the most significant digits. This error is caused by the specific recursive procedure of our choice. Since we assume that the model would easily figure out the difference in the numbers of digits, we did not add explicit steps to count the digits. Although the model detects the difference in most cases, it turns out to be a

relatively challenging operation. A possible solution is to teach the model to explicitly count the number of digits and compare them.

## I  Context Length Distribution

In Figure 4, we present the distributions of context lengths for each problem type. We compare the context lengths of RoT and CoT. For each configuration, we randomly sample 10K contexts from the training distribution and plot the histogram of their lengths. The graphs show that the context sizes of CoT are many orders of magnitude larger than RoT. In theory, the total number of tokens to generate for each problem is identical in both RoT and CoT (if we do not count the `THINK` tokens). However, RoT's context sizes are much smaller since it utilizes multiple contexts.

Another advantage of RoT is the utilization of dynamic programming. Since we can easily cache the duplicate computations of RoT as explained in Appendix G, we can drastically reduce the amount of token generation if there is a redundant structure in the problem. The amount of tokens to generate for each problem is plotted in Figure 5. The benefit is especially prominent in the algorithmic problems. For example, finding the LCS of two 32-digit sequences results in more than $10^{18}$ tokens if we naively use CoT or RoT. If we use dynamic programming with RoT, we can efficiently solve the same problem with much less cost.

## J  Transformers Are Powerful Sorting Machines

In fact, the first algorithmic task that we tested is sorting since it has been widely used as a benchmark for algorithmic reasoning (Reed & de Freitas, 2016; Cai et al., 2017; Pierrot et al., 2019). However, we find that Transformers are incredibly good at sorting, even in the WT setting. Figure 6 shows the sorting experiment. For CoT and RoT, we train the merge sort algorithm. Interestingly, WT easily achieves a perfect score in sorting 64 three-digit numbers. Also the training converges much faster than RoT. The Transformer architecture, more specifically the attention mechanism, seems to be perfectly suited for the sorting operation.

## K  The Exact Values of Figure 3

Table 2-5 show the exact values of the graphs in Figure 3. Except for the GPT-3 experiments in Table 2, we report the average and the standard deviation of eight runs. Each GPT-3 experiment is done only once.

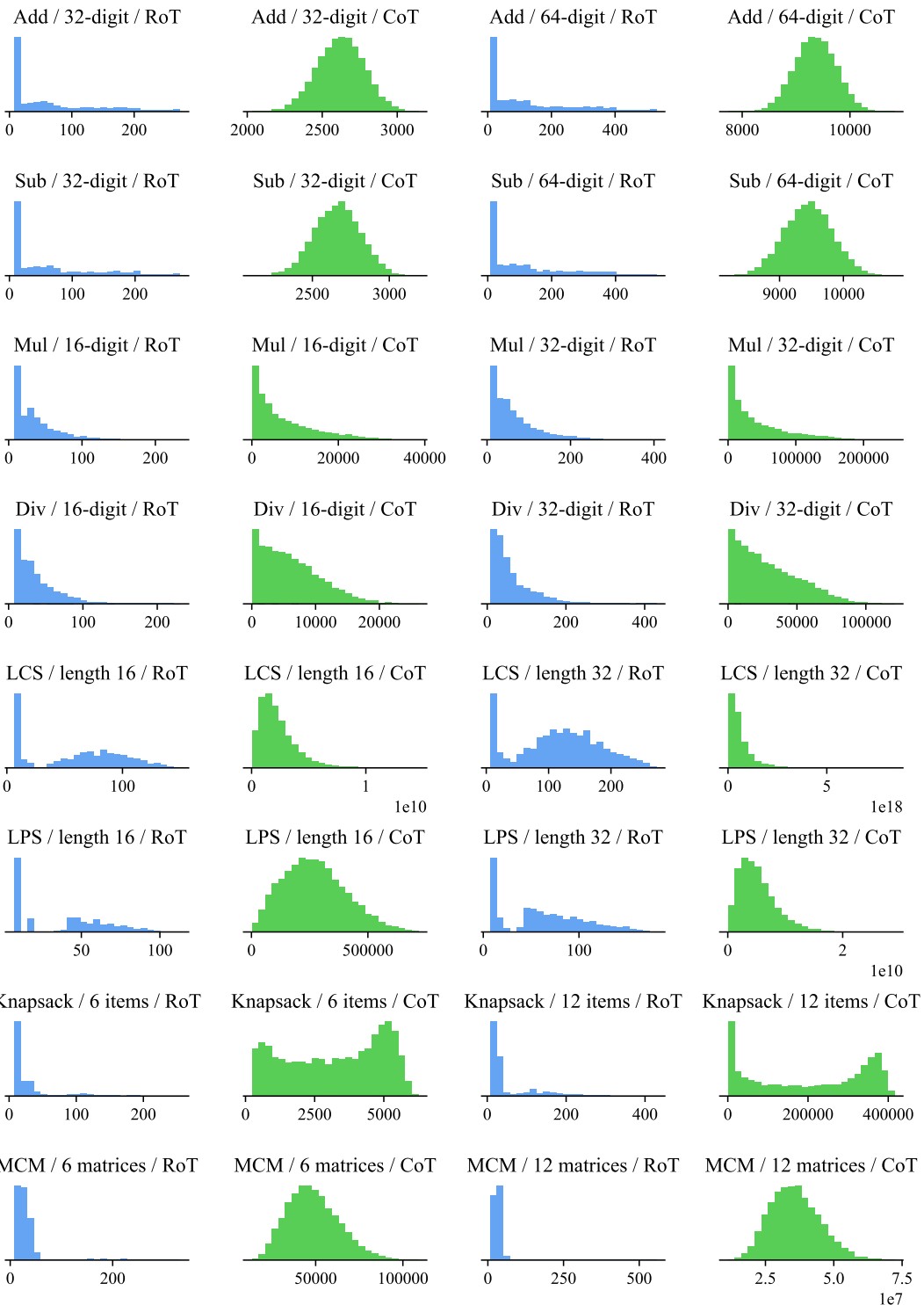

Figure 4: The distributions of context lengths.

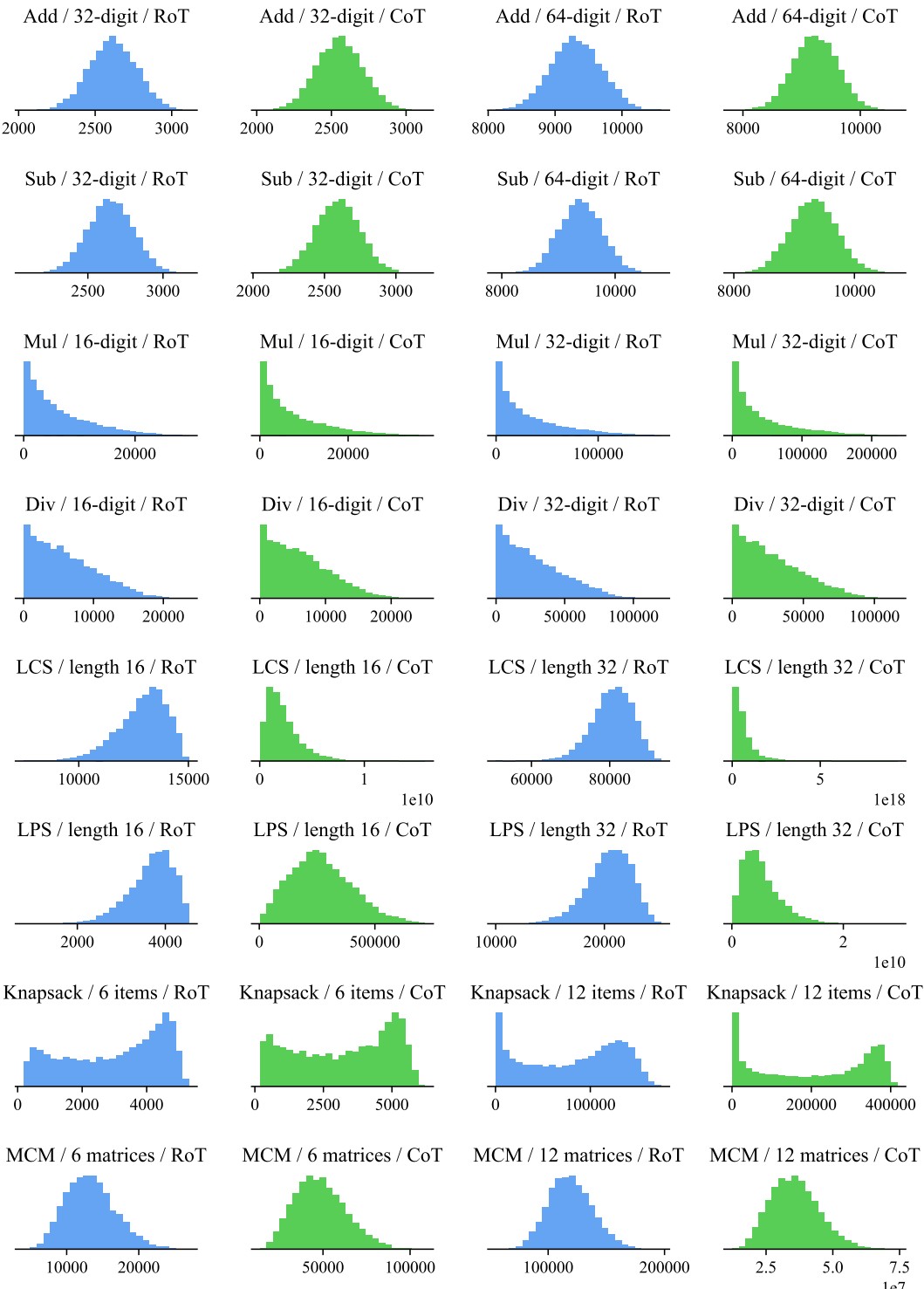

Figure 5: The distribution of the total number of tokens to produce in order to solve each problem. RoT can utilize dynamic programming to reduce redundant computations.

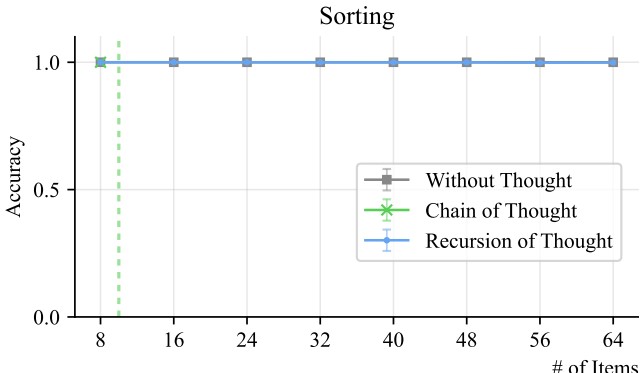

Figure 6: Sorting experiment with the tiny Transformer.

| Problem | Difficulty | WT | CoT | RoT |
|---|---|---|---|---|
| Addition | 32-digit | 0.991 | – | 0.998 |
| | 48-digit | 0.853 | – | 0.995 |
| Subtraction | 32-digit | 0.991 | – | 0.998 |
| | 48-digit | 0.886 | – | 0.998 |
| Multiplication | 8-digit | 0.337 | – | 0.999 |
| | 16-digit | 0.098 | – | 0.994 |
| Division | 8-digit | 0.363 | – | 1.000 |
| | 16-digit | 0.123 | – | 0.989 |
| LCS | length 16 | 0.980 | – | 0.995 |
| | length 24 | 0.832 | – | 0.998 |
| LPS | length 24 | 0.995 | – | 1.000 |
| | length 40 | 0.800 | – | 0.974 |
| 0-1 Knapsack | 4 items | 0.945 | – | 0.999 |
| | 6 items | 0.634 | – | 1.000 |
| MCM | 3 matrices | 0.481 | – | 0.997 |
| | 4 matrices | 0.110 | – | 0.992 |

Table 2: The exact values of the GPT-3 experiments in Figure 3a.

| Problem | Difficulty | WT | CoT | RoT |
|---|---|---|---|---|
| Addition | 8-digit | $0.863 \pm 0.265$ | $1.000 \pm 0.000$ | $1.000 \pm 0.000$ |
| | 16-digit | $0.370 \pm 0.475$ | $1.000 \pm 0.000$ | $1.000 \pm 0.000$ |
| | 24-digit | $0.336 \pm 0.430$ | $1.000 \pm 0.000$ | $1.000 \pm 0.000$ |
| | 32-digit | $0.455 \pm 0.458$ | $-$ | $1.000 \pm 0.000$ |
| | 40-digit | $0.119 \pm 0.316$ | $-$ | $1.000 \pm 0.000$ |
| | 48-digit | $0.082 \pm 0.216$ | $-$ | $1.000 \pm 0.000$ |
| | 56-digit | $0.105 \pm 0.277$ | $-$ | $1.000 \pm 0.000$ |
| | 64-digit | $0.000 \pm 0.000$ | $-$ | $1.000 \pm 0.001$ |
| Subtraction | 8-digit | $0.982 \pm 0.006$ | $1.000 \pm 0.000$ | $1.000 \pm 0.000$ |
| | 16-digit | $0.705 \pm 0.411$ | $1.000 \pm 0.000$ | $1.000 \pm 0.000$ |
| | 24-digit | $0.238 \pm 0.412$ | $1.000 \pm 0.000$ | $1.000 \pm 0.000$ |
| | 32-digit | $0.221 \pm 0.385$ | $-$ | $1.000 \pm 0.000$ |
| | 40-digit | $0.426 \pm 0.433$ | $-$ | $1.000 \pm 0.000$ |
| | 48-digit | $0.114 \pm 0.303$ | $-$ | $1.000 \pm 0.000$ |
| | 56-digit | $0.116 \pm 0.307$ | $-$ | $1.000 \pm 0.000$ |
| | 64-digit | $0.161 \pm 0.282$ | $-$ | $1.000 \pm 0.000$ |
| Multiplication | 2-digit | $1.000 \pm 0.000$ | $1.000 \pm 0.000$ | $1.000 \pm 0.000$ |
| | 4-digit | $0.817 \pm 0.023$ | $1.000 \pm 0.000$ | $1.000 \pm 0.000$ |
| | 8-digit | $0.340 \pm 0.032$ | $-$ | $1.000 \pm 0.000$ |
| | 12-digit | $0.169 \pm 0.015$ | $-$ | $1.000 \pm 0.000$ |
| | 16-digit | $0.104 \pm 0.016$ | $-$ | $1.000 \pm 0.000$ |
| | 20-digit | $0.048 \pm 0.020$ | $-$ | $1.000 \pm 0.000$ |
| | 24-digit | $0.033 \pm 0.017$ | $-$ | $0.999 \pm 0.001$ |
| | 28-digit | $0.014 \pm 0.006$ | $-$ | $0.999 \pm 0.001$ |
| | 32-digit | $0.012 \pm 0.001$ | $-$ | $0.999 \pm 0.000$ |
| Division | 2-digit | $1.000 \pm 0.000$ | $1.000 \pm 0.000$ | $1.000 \pm 0.000$ |
| | 4-digit | $0.978 \pm 0.008$ | $1.000 \pm 0.000$ | $1.000 \pm 0.000$ |
| | 8-digit | $0.354 \pm 0.029$ | $-$ | $1.000 \pm 0.000$ |
| | 12-digit | $0.186 \pm 0.009$ | $-$ | $1.000 \pm 0.000$ |
| | 16-digit | $0.128 \pm 0.011$ | $-$ | $1.000 \pm 0.000$ |
| | 20-digit | $0.087 \pm 0.012$ | $-$ | $1.000 \pm 0.000$ |
| | 24-digit | $0.075 \pm 0.005$ | $-$ | $1.000 \pm 0.000$ |
| | 28-digit | $0.059 \pm 0.007$ | $-$ | $0.999 \pm 0.000$ |
| | 32-digit | $0.048 \pm 0.008$ | $-$ | $0.999 \pm 0.000$ |

Table 3: The exact values of the Transformer experiments in Figure 3b (arithmetic problems).

| Problem | Difficulty | WT | CoT | RoT |
|---|---|---|---|---|
| LCS | length 3 | $1.000 \pm 0.000$ | $1.000 \pm 0.000$ | − |
| | length 4 | $0.997 \pm 0.008$ | − | $1.000 \pm 0.000$ |
| | length 8 | $0.999 \pm 0.002$ | − | $1.000 \pm 0.000$ |
| | length 12 | $0.965 \pm 0.025$ | − | $1.000 \pm 0.000$ |
| | length 16 | $0.880 \pm 0.035$ | − | $1.000 \pm 0.000$ |
| | length 20 | $0.759 \pm 0.043$ | − | $1.000 \pm 0.000$ |
| | length 24 | $0.622 \pm 0.038$ | − | $1.000 \pm 0.000$ |
| | length 28 | $0.484 \pm 0.043$ | − | $0.999 \pm 0.000$ |
| | length 32 | $0.375 \pm 0.030$ | − | $0.999 \pm 0.000$ |
| LPS | length 4 | $1.000 \pm 0.000$ | $1.000 \pm 0.000$ | − |
| | length 7 | $1.000 \pm 0.000$ | $1.000 \pm 0.000$ | − |
| | length 8 | $1.000 \pm 0.000$ | − | $1.000 \pm 0.000$ |
| | length 16 | $0.999 \pm 0.001$ | − | $1.000 \pm 0.000$ |
| | length 24 | $0.950 \pm 0.019$ | − | $1.000 \pm 0.000$ |
| | length 32 | $0.788 \pm 0.019$ | − | $1.000 \pm 0.000$ |
| | length 40 | $0.608 \pm 0.023$ | − | $1.000 \pm 0.000$ |
| | length 48 | $0.477 \pm 0.030$ | − | $0.999 \pm 0.001$ |
| | length 56 | $0.365 \pm 0.029$ | − | $0.998 \pm 0.000$ |
| 0-1 Knapsack | 2 items | $1.000 \pm 0.000$ | $1.000 \pm 0.000$ | $1.000 \pm 0.000$ |
| | 4 items | $0.966 \pm 0.006$ | $1.000 \pm 0.000$ | $1.000 \pm 0.000$ |
| | 6 items | $0.849 \pm 0.007$ | − | $1.000 \pm 0.000$ |
| | 8 items | $0.640 \pm 0.242$ | − | $1.000 \pm 0.000$ |
| | 10 items | $0.481 \pm 0.279$ | − | $1.000 \pm 0.000$ |
| | 12 items | $0.435 \pm 0.252$ | − | $0.988 \pm 0.029$ |
| MCM | 2 matrices | $0.973 \pm 0.009$ | $1.000 \pm 0.000$ | $1.000 \pm 0.000$ |
| | 4 matrices | $0.177 \pm 0.069$ | − | $1.000 \pm 0.000$ |
| | 6 matrices | $0.088 \pm 0.029$ | − | $1.000 \pm 0.000$ |
| | 8 matrices | $0.033 \pm 0.025$ | − | $1.000 \pm 0.000$ |
| | 10 matrices | $0.051 \pm 0.032$ | − | $0.998 \pm 0.001$ |
| | 12 matrices | $0.026 \pm 0.011$ | − | $0.996 \pm 0.002$ |

Table 4: The exact values of the Transformer experiments in Figure 3b (algorithmic problems).

| Problem | Difficulty | WT | CoT | RoT |
|---|---|---|---|---|
| Addition | 2-digit | $1.000 \pm 0.000$ | $1.000 \pm 0.000$ | $1.000 \pm 0.000$ |
| | 4-digit | $0.642 \pm 0.305$ | $1.000 \pm 0.001$ | $1.000 \pm 0.000$ |
| | 6-digit | $0.005 \pm 0.008$ | $0.997 \pm 0.005$ | $0.999 \pm 0.000$ |
| | 8-digit | $0.000 \pm 0.000$ | $0.905 \pm 0.155$ | $0.999 \pm 0.001$ |
| | 10-digit | $0.000 \pm 0.000$ | $0.795 \pm 0.341$ | $0.986 \pm 0.024$ |
| | 12-digit | $0.000 \pm 0.000$ | $-$ | $0.871 \pm 0.275$ |
| | 14-digit | $0.000 \pm 0.000$ | $-$ | $0.358 \pm 0.430$ |
| | 16-digit | $0.000 \pm 0.000$ | $-$ | $0.120 \pm 0.202$ |
| Subtraction | 2-digit | $1.000 \pm 0.000$ | $1.000 \pm 0.000$ | $1.000 \pm 0.000$ |
| | 4-digit | $0.776 \pm 0.179$ | $1.000 \pm 0.000$ | $1.000 \pm 0.000$ |
| | 6-digit | $0.006 \pm 0.001$ | $1.000 \pm 0.000$ | $1.000 \pm 0.000$ |
| | 8-digit | $0.000 \pm 0.000$ | $0.896 \pm 0.252$ | $0.994 \pm 0.016$ |
| | 10-digit | $0.000 \pm 0.000$ | $0.443 \pm 0.377$ | $0.908 \pm 0.236$ |
| | 12-digit | $0.000 \pm 0.000$ | $-$ | $0.507 \pm 0.398$ |
| | 14-digit | $0.000 \pm 0.000$ | $-$ | $0.295 \pm 0.406$ |
| | 16-digit | $0.000 \pm 0.000$ | $-$ | $0.101 \pm 0.137$ |
| Multiplication | 2-digit | $1.000 \pm 0.000$ | $1.000 \pm 0.000$ | $1.000 \pm 0.000$ |
| | 3-digit | $0.855 \pm 0.044$ | $-$ | $1.000 \pm 0.000$ |
| | 4-digit | $0.636 \pm 0.061$ | $-$ | $1.000 \pm 0.000$ |
| | 5-digit | $0.338 \pm 0.063$ | $-$ | $1.000 \pm 0.000$ |
| | 6-digit | $0.270 \pm 0.030$ | $-$ | $0.987 \pm 0.008$ |
| | 7-digit | $0.162 \pm 0.025$ | $-$ | $0.896 \pm 0.105$ |
| | 8-digit | $0.138 \pm 0.025$ | $-$ | $0.670 \pm 0.208$ |
| Division | 1-digit | $1.000 \pm 0.000$ | $1.000 \pm 0.000$ | $1.000 \pm 0.000$ |
| | 2-digit | $1.000 \pm 0.000$ | $-$ | $1.000 \pm 0.000$ |
| | 3-digit | $1.000 \pm 0.001$ | $-$ | $1.000 \pm 0.000$ |
| | 4-digit | $0.891 \pm 0.072$ | $-$ | $1.000 \pm 0.000$ |
| | 5-digit | $0.516 \pm 0.077$ | $-$ | $0.998 \pm 0.004$ |
| | 6-digit | $0.308 \pm 0.069$ | $-$ | $0.996 \pm 0.007$ |
| | 7-digit | $0.192 \pm 0.028$ | $-$ | $0.958 \pm 0.036$ |
| | 8-digit | $0.115 \pm 0.015$ | $-$ | $0.914 \pm 0.090$ |

Table 5: The exact values of the LSTM experiments in Figure 3c.

