# OpenReview forum: "Recursion of Thought: Divide and Conquer Reasoning with Language Models"
_ICLR.cc/2023/Conference — Submitted to ICLR 2023_

### Official Review · Reviewer_kKKS · 2022-10-23

**Confidence:** 3
**Correctness:** 4
**Technical Novelty And Significance:** 4
**Empirical Novelty And Significance:** 3
**Recommendation:** 8

**Clarity, Quality, Novelty And Reproducibility:**

This work has a good and clear presentation of its idea, its specific methodology, and its experiment settings. The quality and originality of the work should meet or exceed the conference standard. My main concern is on how to properly train the RoT framework as it requires people to feed it with hand-designed subproblems. Therefore, it would be great if the authors could explain the limitation of the problems that RoT could solve and how to design proper subproblems in order to use RoT to train the models.

**Strength And Weaknesses:**

Strength:
-

[+] The proposed framework (RoT) is novel and well-motivated.

[+] The work has conducted multiple experiments which demonstrate the great power of RoT on much more complex arithmetic and algorithmic reasoning problems.

Weakness:
-

[-] The training of RoT requires (non-trivial) human inputs to design proper subproblems.

More specifically, I wonder

1) what is the limitation of the problems that RoT could solve: for example,

- Does RoT have the ability to learn backtrack? Some problems may require people to modify the earlier part of the answer based on the new observed information.

- Does RoT have the ability to learn a problem where its subproblems have different structures (as recursion usually requires the same structure in the subproblems)?

- What is the performance of RoT on (maybe small-scale) NP-hard algorithmic problems that may not have the divide-and-conquer structure such as TSP？

2) how to design proper subproblems in order to use RoT to train the models?

- What are the criteria of the subproblems for RoT? What kinds of structures do they need to have？

- Are there systematic approaches to design subproblems for reasoning tasks in general?


**Summary Of The Paper:**

This work proposed a model-agnostic framework, Recursion of Thought (RoT), to release the capacity constraint by the maximum size of a single context in language models. RoT teaches a language model to divide and conquer complex problems by recursively creating multiple contexts; therefore, a complex problem could be solved by utilizing multiple contexts. This work also conducts extensive experiments on arithmetic and algorithmic reasoning tasks to show the power of RoT in helping language models solve problems that require hundreds of thousands of tokens.

Overall, the proposed framework (RoT) is novel and well-motivated, and the authors have conducted multiple experiments which demonstrate the great power of RoT on much more complex arithmetic and algorithmic reasoning problems. One important point missing from this work is how to systematically construct subproblems given any problem in order to teach the model learn the recursion of thoughts. Still, I think this work has great potential to allow language models to achieve better reasoning abilities.

**Summary Of The Review:**

This work proposed a novel model-agnostic framework, Recursion of Thought (RoT), to release the capacity constraint by the maximum size of a single context in language models. RoT teaches a language model to divide and conquer complex problems by recursively creating multiple contexts; therefore, a complex problem could be solved by utilizing multiple contexts. It then conducts extensive experiments on arithmetic and algorithmic reasoning tasks to show the power of RoT in helping language models solve problems that require hundreds of thousands of tokens.

The proposed framework (RoT) is novel and well-motivated, and this work has a good and clear presentation of its idea, its specific methodology, and its experiment settings. The quality and originality of the work should meet or exceed the conference standard. My main concern is on how to properly train the RoT framework as the training of RoT requires (non-trivial) human inputs to design proper subproblems. Still, I think this work has great potential to allow language models to achieve better reasoning abilities.

---

> ### Author Response · Authors · 2022-11-18
> **Reply to Reviewer kKKS**
>
> Thank you for the kind and insightful review.
>
>
> ## What is the limitation of the problems that RoT could solve?
>
> This is an important question.
> At least theoretically, RoT can handle any type of problem as long as it can be formulated as recursion.
> Recursion is extremely general, such that many Turing-complete functional programming languages rely on it without any construct for loops.
> Moreover, the [continuation-passing style](https://en.wikipedia.org/wiki/Continuation-passing_style) enables any function to be represented as a tail-recursive form.
> Therefore, in theory, there is almost no limitation on the problems that RoT can solve.
>
> Note that RoT does not require the subproblems to have the same structures.
> In our experiments, addition appears as a subproblem of multiplication, and both addition and multiplication appear as subproblems of MCM.
> These problems have very different structures, and RoT has no problem handling them.
>
> If we take things to the extreme (e.g., multiplying numbers with hundreds of digits), the maximum size of a single context would be a major constraint.
> Even if we use RoT, we should be able to divide a problem such that the context length is always kept under the limit.
>
> In a practical sense, however, whether RoT can learn and solve a problem *efficiently* is a different matter.
> Since RoT and language models do not work like computers, some types of problems would be relatively challenging to learn for RoT.
> Backtracking would be one such example; since language models cannot modify previously generated tokens, we should teach backtracking in a less efficient way.
>
> About the last bullet point, we believe RoT can also handle NP-hard problems with small enough sizes.
> However, we are not sure if there is a problem that cannot be solved by divide and conquer.
> For TSP, there is a [recursive, exponential-time dynamic programming algorithm](https://www.geeksforgeeks.org/travelling-salesman-problem-set-1/) that follows the principle of divide and conquer.
>
>
> ## How to design proper subproblems
>
> We believe this is an open-ended problem.
> The ideal form of subproblems would vary depending on multiple factors, including the model architecture.
> Even in the case of algorithms for computers, the optimal algorithm varies depending on the processor architecture or available memory space.
> We think it would not be feasible to devise a general, systematic methodology to design subproblems.
> Therefore, we simply choose popular problem decompositions that humans have developed, and interestingly, they seem to work well for RoT.

---

> > ### Comment · Reviewer_kKKS · 2022-11-24
> > **Thank you for the response!**
> >
> > After reading through the other reviews as well as the authors' responses, I still think this paper is of good quality as it well tackled one fundamental problem --- the length limit of a single context. I still recommend this paper be accepted.

---

### Official Review · Reviewer_U3UZ · 2022-10-24

**Confidence:** 5
**Correctness:** 3
**Technical Novelty And Significance:** 2
**Empirical Novelty And Significance:** 1
**Recommendation:** 1

**Clarity, Quality, Novelty And Reproducibility:**

The paper is written clearly.

The novelty is minimal as the solution proposed is rather trivial as it doesn't try to solve the hard problem of reasoning without direct supervision, or other settings such as few-shot/zero-shot/multi-task and formulates the problem as a fully supervised task.

Due to simplicity of approach, I think it's reproducible.

**Strength And Weaknesses:**

On the positive side, the paper is well written and easy to read.

On the negative side, the solution proposed is formulated as a fully supervised problem, meaning for each task the exact steps needed to solve the problem needs to be annotated. This is not a scalable approach as for realistic real-world applications obtaining detailed annotations for reasoning problems is an expert task which is expensive to scale to large datasets. As a result, outside of tasks such as arithmetic operations where the dataset can be created automatically (and one could argue that you wouldn't use a task specific LM as the paper doesn't deal with multiple tasks or out-of-distribution problems), the applicability of this approach is minimal.

**Summary Of The Paper:**

This paper experiments with fully supervised training of a LM to produce subsequences needed for solving simple reasoning tasks, motivated by limited context size of LMs.

**Summary Of The Review:**

Unfortunately, I don't see much impact from this work in practical terms. The formulation doesn't scale to general reasoning problems, the tasks considered aren't tasks that you would train a specific LM to solve, and the algorithmic solution is very simple/trivial extension. Hence, I'm not able to recommend this paper for publication.

---

> ### Author Response · Authors · 2022-11-18
> **Reply to Reviewer U3UZ**
>
> Thank you for your review.
> Here we summarize our novelty and provide explanations for your criticisms.
> Thank you for taking time to review our paper, and we hope you reassess its value.
>
>
> ## The Multi-Context Paradigm
>
> The main novelty of our work is the introduction of the multi-context paradigm.
> We believe there would be no disagreement on the limitation of relying solely on a single context, which is usually capped at just a few thousand tokens.
> Our work is first to address this issue by utilizing multiple contexts, following the principle of divide and conquer.
> RoT is a novel framework that _tokenizes_ context control operations, and thus it can be combined with any kind of language model.
> The experiments show that RoT makes LMs solve complex problems whose solutions consist of hundreds of thousands of tokens.
>
>
> ## CoT Do Not Scale to General Reasoning Problems
>
> We call attention to the fact that CoT is not a general formulation due to the length limit of a single context.
> The problems in our experiments are the counter-examples proving that CoT cannot be a general reasoning framework.
> We propose RoT as a more general framework that can scale up to far more complex problems than CoT.
>
>
> ## Supervised Learning Is a Good Starting Point
>
> Although we currently adopt supervised training to focus on the multi-context aspect, **it is a common practice to start with strong supervision and then develop more general methods with less supervision.**
>
> NPI (Reed & de Feritas, 2016) is an important example related to symbolic reasoning.
> Being cited hundreds of times, it has inspired a lot of follow-up research.
> However, the original NPI relies on fully-supervised training, similar to RoT.
> As we mentioned in sections 2 and 5, several semi-supervised training techniques for NPI are developed later, and they would be great references for building semi-supervised training schemes for RoT.
> We believe the current situation of RoT is somewhat analogous to NPI.
> Both of them are novel frameworks with a lot of potential, but initially resort to supervised learning.
>
> Scratchpad (Nye et al., 2021) also uses supervised learning, but it is an important early work that shows the benefits of generating intermediate steps.
> CoT prompting (Wei et al., 2022) is developed later, demonstrating that pre-trained language models can be induced to produce intermediate steps once given the right prompts.
> As we summarize below, our work is the first work that shows the benefits of utilizing multiple contexts.
>
>
> | Learning Type | Single-Context | Multi-Context |
> |---------------|----------------|---------------|
> | Supervised | Scratchpad (Nye et al., 2021) | RoT (ours) |
> | Weakly Supervised | CoT prompting (Wei et al., 2022) | ? (future work) |
>
>
>
> ## The Significance of Our Experiments
>
> We chose the eight problem types for our experiments because we can easily increase the complexity to quantitatively show the necessity of the multi-context paradigm.
> We emphasize that we increase the problem complexity to the point where the intermediate steps consist of hundreds of thousands of tokens (see appendix I for the number of tokens).
> To our knowledge, no previous work (including fully supervised ones) comes close to solving this level of complexity.
> Our experiments are significant, not because the tasks are useful, but because they reveal the limitation of existing approaches that RoT can overcome.
>
> Let's take the recent example of Minerva (Lewkowycz et al., 2022) from Google, which is one of the state-of-the-art language models for reasoning.
> Despite its remarkable high-level reasoning capability, it still struggles to solve basic arithmetic tasks if the number of digits becomes more than a few.
> According to the paper, it scores around 80% on 10-digit addition and 20% on 18-digit addition.
> On the other hand, RoT achieves near-perfect accuracy on 64-digit addition and 32-digit multiplication even with the tiny language models which are a million times smaller than Minerva.
> Considering the previous work on reasoning, proving that language models (even tiny ones) are capable of solving this level of complexity with the multi-context paradigm is a significant contribution.

---

### Official Review · Reviewer_k2RA · 2022-10-28

**Confidence:** 4
**Correctness:** 2
**Technical Novelty And Significance:** 2
**Empirical Novelty And Significance:** 2
**Recommendation:** 3

**Clarity, Quality, Novelty And Reproducibility:**

The writing style is a bit "flashy", which gives room for miscommunications. For example, the paper opens with the following statement:

"Although neural networks have achieved amazing results on various domains, e.g., images, texts,
audios, videos, games, etc., nearly all of them are classified as System 1 tasks (Kahneman, 2013), ..."

Afaik, this statement is not supported by the literature, and in particular not by the given reference. Following Kahneman's theory, tasks like playing the board game go would certainly require system 2 thinking. IMO, dropping the first paragraph would help the paper.

The paper also makes claims such as "the length of CoT can grow rapidly with the problem’s complexity", without further explanation and I am not even sure what exactly this means.

**Strength And Weaknesses:**

My main concern with this approach is that the paper does not discuss how to apply this approach to more general problems. The problems considered here are very simple, and an algorithm to solve them needs to be known in order to generate the training data. So, in its current form this does not enable any new abilities as we could use the base algorithm instead of using expensive LMs.

In terms of the insight from a learning point-of-view, I also do not see anything surprising in this work. After decomposing the problem manually (i.e. by writing an algorithm to generate training data), the learning task is pretty simple (for addition, for example, all the model has to do is to extract the last digit from two numbers). So the high accuracy of the resulting method is not that surprising.

**Summary Of The Paper:**

This paper demonstrates that for simple algorithmic problems (arithmetic), language models can be taught to split problems into multiple subproblems, which can then be fed back to the LM to be solved independently. With very small models this can achieve great performance on these tasks, and it circumvents the limit in context-length of existing transformer models.

**Summary Of The Review:**

This paper demonstrates that decomposing simple algorithmic problems can circumvent the limit in context length language models.

However, the technique only seems to work for problems for which we already have algorithms, and it is unclear to me if this could be extended to more general problems. The writing style should be improved.

---

> ### Author Response · Authors · 2022-11-18
> **Reply to Reviewer k2RA (Part 1)**
>
> Thank you for the thoughtful review.
> In the following, we summarize our novelty and elucidate why our paper is valuable despite the concerns.
>
>
> ## The Multi-Context Paradigm
>
> The main novelty of our work is the introduction of the multi-context paradigm.
> We believe there would be no disagreement on the limitation of relying solely on a single context, which is usually capped at just a few thousand tokens.
> Our work is first to address this issue by utilizing multiple contexts, following the principle of divide and conquer.
> RoT is a novel framework that _tokenizes_ context control operations, and thus it can be combined with any kind of language model.
> The experiments show that RoT makes LMs solve complex problems whose solutions consist of hundreds of thousands of tokens.
>
>
> ## Clarifying a Potential Misunderstanding
>
> > In terms of the insight from a learning point-of-view, I also do not see anything surprising in this work. After decomposing the problem manually (i.e. by writing an algorithm to generate training data), the learning task is pretty simple (for addition, for example, all the model has to do is to extract the last digit from two numbers).
>
> We think there might be a misunderstanding here.
> We would like to clarify that the algorithms are used to generate training data, and the model is on its own at test time.
> In other words, at test time, there is no help from the algorithms, and the model has already learned to decompose the problems by producing special tokens.
> In the example of addition, the model should learn not only extracting the last digits but also how to decompose the problems with the special tokens.
> To the best of our knowledge, this approach has not been introduced before, and we hope our novelty is reassessed.
>
>
> ## Supervised Learning Is a Good Starting Point
>
> Although we currently adopt supervised training to focus on the multi-context aspect, **it is a common practice to start with strong supervision and then develop more general methods with less supervision.**
>
> NPI (Reed & de Feritas, 2016) is an important example related to symbolic reasoning.
> Being cited hundreds of times, it has inspired a lot of follow-up research.
> However, the original NPI relies on fully-supervised training, similar to RoT.
> As we mentioned in sections 2 and 5, several semi-supervised training techniques for NPI are developed later, and they would be great references for building semi-supervised training schemes for RoT.
> We believe the current situation of RoT is somewhat analogous to NPI.
> Both of them are novel frameworks with a lot of potential, but initially resort to supervised learning.
>
> Scratchpad (Nye et al., 2021) also uses supervised learning, but it is an important early work that shows the benefits of generating intermediate steps.
> CoT prompting (Wei et al., 2022) is developed later, demonstrating that pre-trained language models can be induced to produce intermediate steps once given the right prompts.
> As we summarize below, our work is the first work that shows the benefits of utilizing multiple contexts.
>
>
> | Learning Type | Single-Context | Multi-Context |
> |---------------|----------------|---------------|
> | Supervised | Scratchpad (Nye et al., 2021) | RoT (ours) |
> | Weakly Supervised | CoT prompting (Wei et al., 2022) | ? (future work) |
>
>
> ## The Significance of Our Experiments
>
> We chose the eight problem types for our experiments because we can easily increase the complexity to quantitatively show the necessity of the multi-context paradigm.
> We emphasize that we increase the problem complexity to the point where the intermediate steps consist of hundreds of thousands of tokens (see appendix I for the number of tokens).
> To our knowledge, no previous work (including fully supervised ones) comes close to solving this level of complexity.
> Our experiments are significant, not because the tasks are useful, but because they reveal the limitation of existing approaches that RoT can overcome.
>
> Let's take the recent example of Minerva (Lewkowycz et al., 2022) from Google, which is one of the state-of-the-art language models for reasoning.
> Despite its remarkable high-level reasoning capability, it still struggles to solve basic arithmetic tasks if the number of digits becomes more than a few.
> According to the paper, it scores around 80% on 10-digit addition and 20% on 18-digit addition.
> On the other hand, RoT achieves near-perfect accuracy on 64-digit addition and 32-digit multiplication even with the tiny language models which are a million times smaller than Minerva.
> Considering the previous work on reasoning, proving that language models (even tiny ones) are capable of solving this level of complexity with the multi-context paradigm is a significant contribution.

---

> ### Author Response · Authors · 2022-11-18
> **Reply to Reviewer k2RA (Part 2)**
>
> ## Exploring a Fundamental Aspect of Reasoning
>
> Like many other contemporary works, this paper is a groundwork for future language models, rather than a complete solution for reasoning.
> Numerous open problems should be solved to achieve human-level reasoning, and we focus on one fundamental problem: the length limit of a single context.
> This is why we adopt several simplified assumptions, such as supervised learning, and deal with problems for which efficient algorithms are already known.
> We believe the significance of using multiple contexts substantially outweighs the current limitations of RoT since it would be a necessary condition to solve reasoning.
> We would be sincerely grateful if you reconsider your decision.
>
>
> ## Paper Updates
>
> - We removed the first paragraph.
> - We rewrote the paragraph containing "the length of CoT can grow rapidly with the problem’s complexity."

---

### Decision · Program_Chairs · 2023-01-20

**Decision:**

Reject

**Justification For Why Not Higher Score:**

The novelty of the paper is below expectation. see meta review for details.

**Justification For Why Not Lower Score:**

The vote is Rejection.

**Metareview: Summary, Strengths And Weaknesses:**

The paper considers chain of thought approach to prompting and the problem of limited context length in current LLMs. The authors propose to use recursion to deal with limited context length and provide experiments on arithmetic and reasoning tasks to show the performance of the proposed approach.

Strength:
- The paper is well-written and easy to follow.
- Limited context length is an important issue for the use of current LLMs.


Weaknesses:
- The idea of using recursion (and splitting into subproblems) to deal with limited context length in LLMs is not new and has been used for other settings such as mathematical theorem proving.
- The authors make general claims about their proposed approach without the idea being general enough as it is hand crafted.

As is, the paper is not ready for publication.